# Single-cell transcriptomes and whole-brain projections of serotonin neurons in the mouse dorsal and median raphe nuclei

Jing Ren[1†]*, Alina Isakova[2,3†], Drew Friedmann[1†], Jiawei Zeng[4†], Sophie M Grutzner[1], Albert Pun[1], Grace Q Zhao[5], Sai Saroja Kolluru[2,3], Ruiyu Wang[4], Rui Lin[4], Pengcheng Li[6,7], Anan Li[6,7], Jennifer L Raymond[5], Qingming Luo[6], Minmin Luo[4,8], Stephen R Quake[2,3,9]*, Liqun Luo[1]*

[1]Department of Biology and Howard Hughes Medical Institute, Stanford University, Stanford, United States; [2]Department of Bioengineering, Stanford University, Stanford, United States; [3]Department of Applied Physics, Stanford University, Stanford, United States; [4]National Institute of Biological Science, Beijing, China; [5]Department of Neurobiology, Stanford University School of Medicine, Stanford, United States; [6]Britton Chance Center for Biomedical Photonics, Wuhan National Laboratory for Optoelectronics, Huazhong University of Science and Technology (HUST), Wuhan, China; [7]HUST-Suzhou Institute for Brainsmatics, JITRI Institute for Brainsmatics, Suzhou, China; [8]School of Life Science, Tsinghua University, Beijing, China; [9]Chan Zuckerberg Biohub, San Francisco, United States

*For correspondence:
jren@mrc-lmb.cam.ac.uk (JR);
steve@quake-lab.org (SRQ);
lluo@stanford.edu (LL)

†These authors contributed equally to this work

**Abstract** Serotonin neurons of the dorsal and median raphe nuclei (DR, MR) collectively innervate the entire forebrain and midbrain, modulating diverse physiology and behavior. To gain a fundamental understanding of their molecular heterogeneity, we used plate-based single-cell RNA-sequencing to generate a comprehensive dataset comprising eleven transcriptomically distinct serotonin neuron clusters. Systematic in situ hybridization mapped specific clusters to the principal DR, caudal DR, or MR. These transcriptomic clusters differentially express a rich repertoire of neuropeptides, receptors, ion channels, and transcription factors. We generated novel intersectional viral-genetic tools to access specific subpopulations. Whole-brain axonal projection mapping revealed that DR serotonin neurons co-expressing vesicular glutamate transporter-3 preferentially innervate the cortex, whereas those co-expressing thyrotropin-releasing hormone innervate subcortical regions in particular the hypothalamus. Reconstruction of 50 individual DR serotonin neurons revealed diverse and segregated axonal projection patterns at the single-cell level. Together, these results provide a molecular foundation of the heterogenous serotonin neuronal phenotypes.

DOI: https://doi.org/10.7554/eLife.49424.001

## Introduction

Serotonin is a phylogenetically ancient signaling molecule (*Hay-Schmidt, 2000*) and the most widely distributed neuromodulator in the brain (*Dahlström and Fuxe, 1964*; *Steinbusch, 1981*). The serotonin system innervates nearly every region of the brain (*Jacobs and Azmitia, 1992*), even though it only constitutes ~1/200,000 of all CNS neurons in humans. It is critically involved in a broad range of brain functions and is the most frequently targeted neural system pharmacologically for treating psychiatric disorders (*Belmaker and Agam, 2008*; *Ravindran and Stein, 2010*).

Serotonin neurons in the central nervous system are spatially clustered in the brainstem, originally designated as groups B1–B9 (*Dahlström and Fuxe, 1964*). Groups B1–B3 are located in the medulla and provide descending serotonergic innervation to the spinal cord and other parts of the medulla. Groups B4–B9 are located in the pons and midbrain, including the dorsal raphe (DR; groups B6 and B7) and median raphe (MR; groups B5 and B8) nucleus, and provide ascending innervation to the forebrain and midbrain. The DR and MR serotonin systems have been linked with the regulation of many mental states and processes, including anxiety, mood, impulsivity, aggression, learning, reward, social interaction, and hence remain the focus of intense research (*Deneris and Wyler, 2012*; *Olivier, 2015*).

Evidence has suggested that the DR and MR serotonin systems differ in developmental origin, connectivity, physiology, and behavioral function (*Alonso et al., 2013*; *Calizo et al., 2011*; *Okaty et al., 2019*). DR serotonin neurons derive entirely from rhombomere 1 of the developing mouse brain, whereas MR serotonin neurons derive predominantly from rhombomeres 1, 2, and 3 (*Alonso et al., 2013*; *Bang et al., 2012*; *Jensen et al., 2008*). Although the DR and MR receive similar inputs globally from specific brain regions (*Ogawa et al., 2014*; *Pollak Dorocic et al., 2014*; *Weissbourd et al., 2014*), they project to largely complementary forebrain targets. The MR serotonin neurons project to structures near the midline, whereas the DR serotonin neurons target more lateral regions (*Jacobs and Azmitia, 1992*; *Muzerelle et al., 2016*; *Vertes, 1991*; *Vertes et al., 1999*). Slice physiology recording showed that the serotonin neurons in the MR and DR have different electrophysiological characteristics, such as resting potential, membrane resistance, and reaction to serotonin receptor-1A agonist (*Calizo et al., 2011*; *Fernandez et al., 2016*). Finally, activation of these two raphe nuclei has been suggested to mediate opposing roles in emotional regulation (*Teissier et al., 2015*).

Even within the MR or DR, there is considerable heterogeneity of serotonin neurons in multiple aspects. Although MR serotonin neurons arising from different cell lineages are anatomically mixed in the adult, they have distinct electrophysiological properties (*Okaty et al., 2015*) and potentially distinct behavioral functions (*Kim et al., 2009*; *Okaty et al., 2015*). Diversity of serotonin neurons in the DR has received particular attention in recent years. Accumulating evidence indicates that there are subgroup-specific projection patterns within the DR serotonin system (*Niederkofler et al., 2016*; *Ren et al., 2018*). The electrophysiological properties of DR serotonin neurons vary according to the projection patterns (*Fernandez et al., 2016*). Physiological recordings as well as optogenetic and chemogenetic manipulations suggest heterogeneity of DR serotonin neurons in their behavioral functions (*Cohen et al., 2015*; *Marcinkiewcz et al., 2016*; *Niederkofler et al., 2016*). As a specific example, we recently found that DR serotonin neurons that project to frontal cortex and amygdala constitute two sub-systems with distinct cell body locations, axonal collateralization patterns, biased inputs, physiological response properties, and behavioral functions (*Ren et al., 2018*). Our collateralization analyses also imply that there must be additional parallel sub-systems of DR serotonin neurons that project to brain regions not visited by the frontal cortex- and amygdala-projecting sub-systems.

Ultimately, the heterogeneity of DR and MR serotonin neurons must be reflected at the molecular level. Pioneering work has introduced the molecular diversity of serotonin neurons across the midbrain and hindbrain (*Okaty et al., 2015*; *Spaethling et al., 2014*; *Wylie et al., 2010*), yet systematic analysis and integration of multiple cellular characteristics at single-cell resolution within each raphe nucleus is still lacking. The rapid development of single-cell RNA sequencing (scRNA-seq) technology in recent years has provided a powerful tool for unbiased identification of transcriptomic cell types in the brain (*Darmanis et al., 2015*; *Li et al., 2017*; *Mickelsen et al., 2019*; *Rosenberg et al., 2018*; *Saunders et al., 2018*; *Tasic et al., 2016*; *Tasic et al., 2018*; *Welch et al., 2019*; *Zeisel et al., 2018*; *Zeisel et al., 2015*). In neural circuits where cell types have been well studied by anatomical and physiological methods, there is an excellent correspondence between cell types defined by transcriptomes and the classical methods (*Li et al., 2017*; *Shekhar et al., 2016*). Here, we combine scRNA-seq, fluorescence in situ hybridization, intersectional labeling of genetically defined cell types, whole-brain axonal projection mapping, and single neuron reconstruction to investigate the relationship between the molecular architecture of serotonin neurons, the spatial location of their cell bodies in the DR and MR, and their axonal arborization patterns in the brain.

## Results

### Single-cell RNA-sequencing defines 11 transcriptomic clusters of serotonin neurons in the dorsal and median raphe

We performed a comprehensive survey of DR and MR serotonin neurons in the adult mouse brain by scRNA-seq (*Figure 1A*). To specifically label serotonin neurons, we crossed *Sert-Cre* mice (*Gong et al., 2007*) with the tdTomato Cre reporter mouse, Ai14 (*Madisen et al., 2010*). (Serotonin transporter, or Sert, encoded by the gene *Slc6a4,* is a marker for serotonin neurons; see more details below.) We collected serotonin neurons acutely dissociated from brain slices of adult mice (postnatal day 40–45, including six females and eight males) by fluorescence-activated cell sorting (FACS) and used Smart-seq2 (*Picelli et al., 2013*) to generate scRNA-seq libraries. To assist in localizing the resulting subtypes, we applied two dissection strategies to separate the serotonin neurons originating from anatomically-distinct brain regions: 1) in the first set of experiments, we dissected the brainstem region that contain the entire MR and DR; 2) in the second set of experiments, we focused on the principal DR (pDR, corresponding to the traditional B7 group) region by dissecting specifically the DR but excluding its caudal extension (cDR, corresponding to the traditional group B6) (*Figure 1—figure supplement 1A*). After quality control (Materials and methods), we determined the transcriptomes of 709 cells from eight samples that include MR, pDR, and cDR, and 290 cells from six pDR-only samples (999 cells in total, comprising 581 cells from female and 418 cells from male mice). We sequenced to a depth of ~1 million reads per cell and detected ~10,000 genes per cell (*Figure 1—figure supplement 1B*; *Supplementary file 1*). The data quality and serotonin identity were validated by the fact that nearly all 999 neurons expressed the genes encoding: (1) tryptophan hydroxylase 2 (Tph2), a key enzyme for serotonin biosynthesis; (2) transcription factor Pet1 (Fev) known to express in raphe serotonin neurons (*Hendricks et al., 1999*); (3) the plasma membrane serotonin transporter (Sert), which recycles released serotonin back to presynaptic terminals of serotonin neurons, and (4) vesicular monoamine transporter 2 (Vmat2), which transports serotonin (and other monoamines) from presynaptic cytoplasm to synaptic vesicles (*Figure 1—figure supplement 1C*).

To define distinct serotonin neuron populations based on single-cell transcriptome, we performed principal component analysis (PCA) on all the genes expressed in the assayed neurons, followed by nearest-neighbor graph-based clustering (*see* Materials and methods). These 999 MR and DR serotonin neurons comprised 11 clusters (*Figure 1B*; *Figure 1—figure supplement 2*). Each cluster contained cells from both sexes after removing genes located on the Y chromosome, indicating that there were few sex-specific differences at the level of transcriptomic clusters (*Figure 1—figure supplement 1D*). No substantial batch effect was observed (*Figure 1—figure supplement 1E*). Each of the 11 clusters expressed a set of cluster-discriminatory genes, including markers for specific neurotransmitter systems, such as *Vglut3*, *Gad1*, and *Gad2* (*Figure 1C,D*; *Figure 1—figure supplement 1–6*).

### Anatomical organization of transcriptomically defined serotonin clusters

Of the 11 transcriptomic clusters, six (Cluster 1–6) consisted of only serotonin neurons dissected from the pDR. We hypothesized that these six clusters represent serotonin neurons from the pDR, and the remaining five clusters represent cells from MR and cDR. To test this hypothesis and to obtain information about the anatomical organization of transcriptomically defined serotonin cell clusters within the DR and MR, we chose 16 cluster marker genes and performed hybridization-chain reaction (HCR)-based single-molecule fluorescence in situ hybridization (smFISH) (*Choi et al., 2018*). To restrict our analysis within the serotonin neuron population, we simultaneously double-labeled *Tph2*, a marker for serotonin neurons, for all the HCR-smFISH experiments.

*Figure 2A* summarizes the distribution of the *Tph2*$^+$ serotonin neurons that express each of the 16 cluster markers in four coronal sections that cover the pDR, cDR, and MR. This summary was derived from counting Tph2/cluster marker double-labeled cells from confocal sections of the HCR-smFISH experiments (*Figure 2B*; *Figure 2—figure supplements 1–2*). Specifically, we found that the distribution of markers for Clusters 1–6, *Ret*, *Trh*, *Gad1*, *Npas1*, *Syt2*, and *C1ql2* (*Figure 1D*), were mostly restricted to the pDR. The two common Cluster 7 markers *Tacr3* and *Met* were both

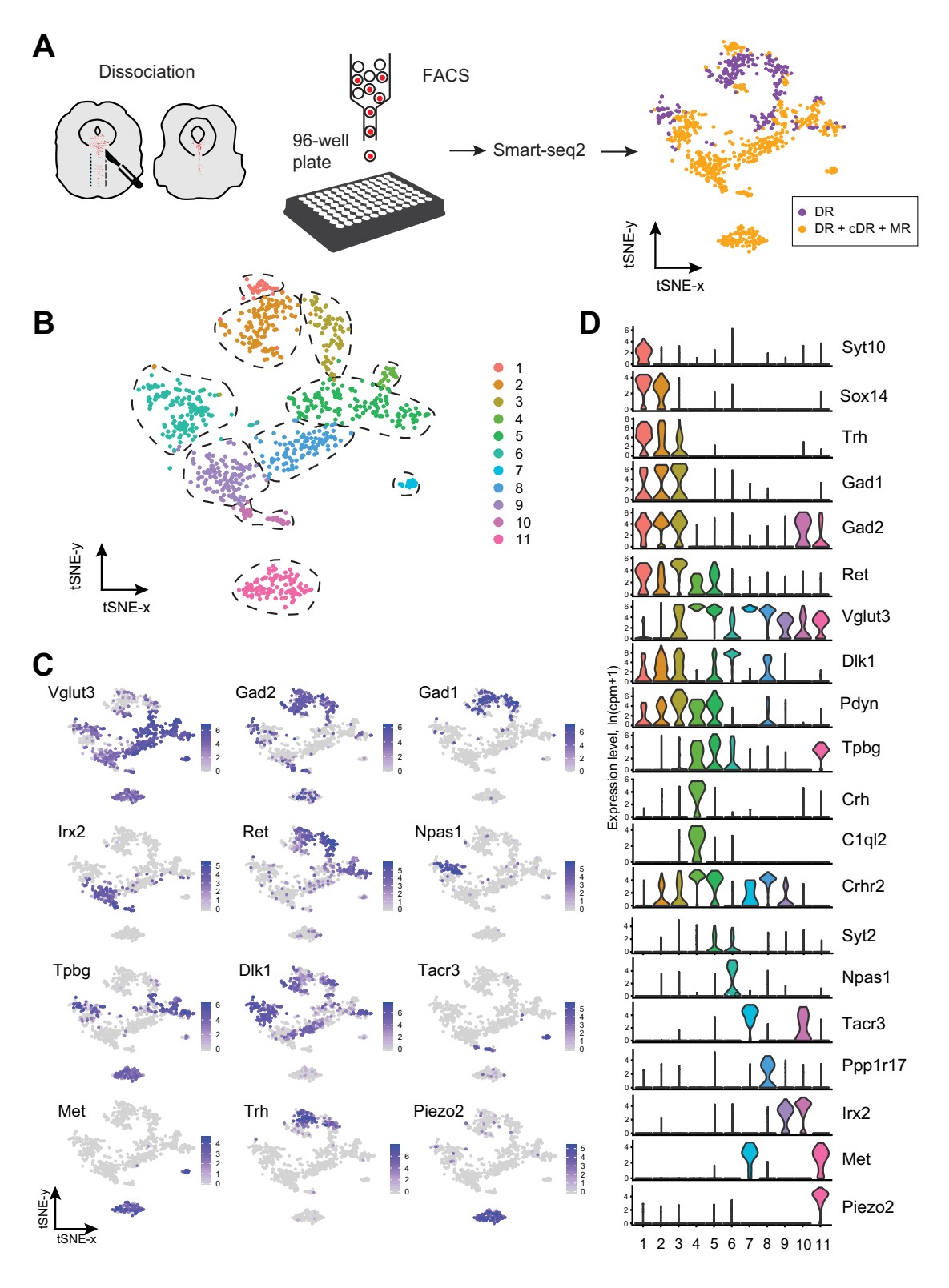

**Figure 1.** Single-cell transcriptomic profiling of serotonergic neurons. (**A**) Schematic representation of scRNA-seq pipeline used to analyze individual serotonin neurons. Tissue slices containing either principal dorsal raphe (pDR, n = 6 brains) or entire DR and MR (n = 8 brains) of *Sert-Cre;Ai14* adult were dissociated to a single-cell suspension. tdTomato⁺ neurons were FACS-sorted in 96-well plates and processed for scRNA-seq using Smart-seq2 protocol. tSNE plot of all processed *Tph2⁺* neurons colored by anatomical localization. cDR, caudal DR; MR, median raphe. (**B**) tSNE plot of 999 *Tph2⁺*

*Figure 1 continued on next page*

*Figure 1 continued*

cells obtained from 14 brains and clustered by gene expression. Cells are colored-coded according to identified transcriptomic clusters. (**C**) Expression of genes defining distinct serotonin neuron populations. Cells are colored by log-normalized expression of each transcript, and the color legend reflects the expression values of ln(CPM+1). CPM, counts per million. (**D**) Violin plots of expression of marker genes across 11 clusters.

DOI: https://doi.org/10.7554/eLife.49424.002

The following figure supplements are available for figure 1:

**Figure supplement 1.** Technical characteristics of scRNA-seq experiments.
DOI: https://doi.org/10.7554/eLife.49424.003

**Figure supplement 2.** Heatmap of gene expression of top ten marker genes identified for each cluster.
DOI: https://doi.org/10.7554/eLife.49424.004

**Figure supplement 3.** Expression patterns of selected genes in DR and MR across individual serotonin neurons presented as tSNE plots (Part I).
DOI: https://doi.org/10.7554/eLife.49424.005

**Figure supplement 4.** Expression patterns of selected genes in DR and MR across individual serotonin neurons presented as tSNE plots (Part II).
DOI: https://doi.org/10.7554/eLife.49424.006

**Figure supplement 5.** Expression patterns of selected genes in DR and MR across individual serotonin neurons presented as tSNE plots (Part III).
DOI: https://doi.org/10.7554/eLife.49424.007

**Figure supplement 6.** Expression patterns of selected genes in DR and MR across individual serotonin neurons presented as tSNE plots (Part IV).
DOI: https://doi.org/10.7554/eLife.49424.008

highly concentrated in serotonin neurons under the aqueduct in the cDR. *Dlk1* should be expressed in the DR clusters as well as Cluster 8 (*Figure 1D*), and its expression was found in both the DR and MR, suggesting that Cluster 8 serotonin neurons are located in the MR. Clusters 9–11 markers *Irx2* and *Piezo2* were mostly found in the MR. Thus, these observations support the anatomical break-down suggested by the dissection of primary tissue, and additionally provide a more granular and detailed description about finer boundaries.

Within the DR, $Trh^+$, $Gad1^+$, and $Gad2^+$ serotonin neurons were mainly located in the dorsal DR, whereas $Slc17a8^+$ (also known as *Vglut3*, used hereafter) and $Syt2^+$ serotonin neurons were mainly located in the ventral DR and cDR. These data suggest that Clusters 1–3 correspond to the dorsal DR and Cluster 4–6 to the ventral DR. Cluster 6 marker $Npas1^+$ was largely excluded from the densest portion of *Tph2* expression at the midline and instead was found scattered in the more rostral and ventral portion of the lateral wings. On the other hand, *Crhr2*, which should be expressed in all DR serotonin neuron clusters except Cluster 1 and 6 (*Figure 1D*), was localized preferentially near the midline and was absent from the lateral wing. Thus, Cluster 6 likely corresponds to serotonin neurons located preferentially in the lateral wings. In contrast to DR, the anatomical organization of the molecular features that define MR clusters is less obvious and different clusters appear more intermingled.

In summary, our HCR-smFISH experiments support the notion that Clusters 1–6 correspond to pDR serotonin neurons, Cluster 7 corresponds to cDR serotonin neurons, and Clusters 8–11 correspond to MR serotonin neurons. We thus rename hereafter Clusters 1–6 as DR-1–6, Cluster 7 as cDR, and Clusters 8–11 as MR-1–4 (*Figure 3*). Detailed expression levels of marker genes across all 11 clusters can be found in *Figure 3—figure supplement 1–3*.

## Molecular properties of MR and DR serotonin neurons

Having determined the spatial locations of transcriptomically defined serotonin cell types, we next analyzed key groups of differentially expressed genes crucial for neuronal function, including markers for neurotransmitter systems, neuropeptides, ionotropic and metabotropic (G-protein-coupled) neurotransmitter receptors, wiring specificity molecules, and transcription factors (*Figure 4*; *Figure 4—figure supplement 1*, *Supplementary file 2*).

### Genes related to neurotransmitters other than serotonin

The majority of the clusters express *Slc17a8*, which encodes vesicular glutamate transporter-3 (Vglut3). These include almost all serotonin neurons from the cDR, the majority from MR clusters (*Domonkos et al., 2016*), and DR-3–6 clusters (*Figure 1D*). These observations suggest that glutamate is the most prevalent co-transmitter for serotonin neurons. Glutamate co-release from $Vglut3^+$ serotonin terminals has indeed been reported at the hippocampus (*Varga et al., 2009*), orbital

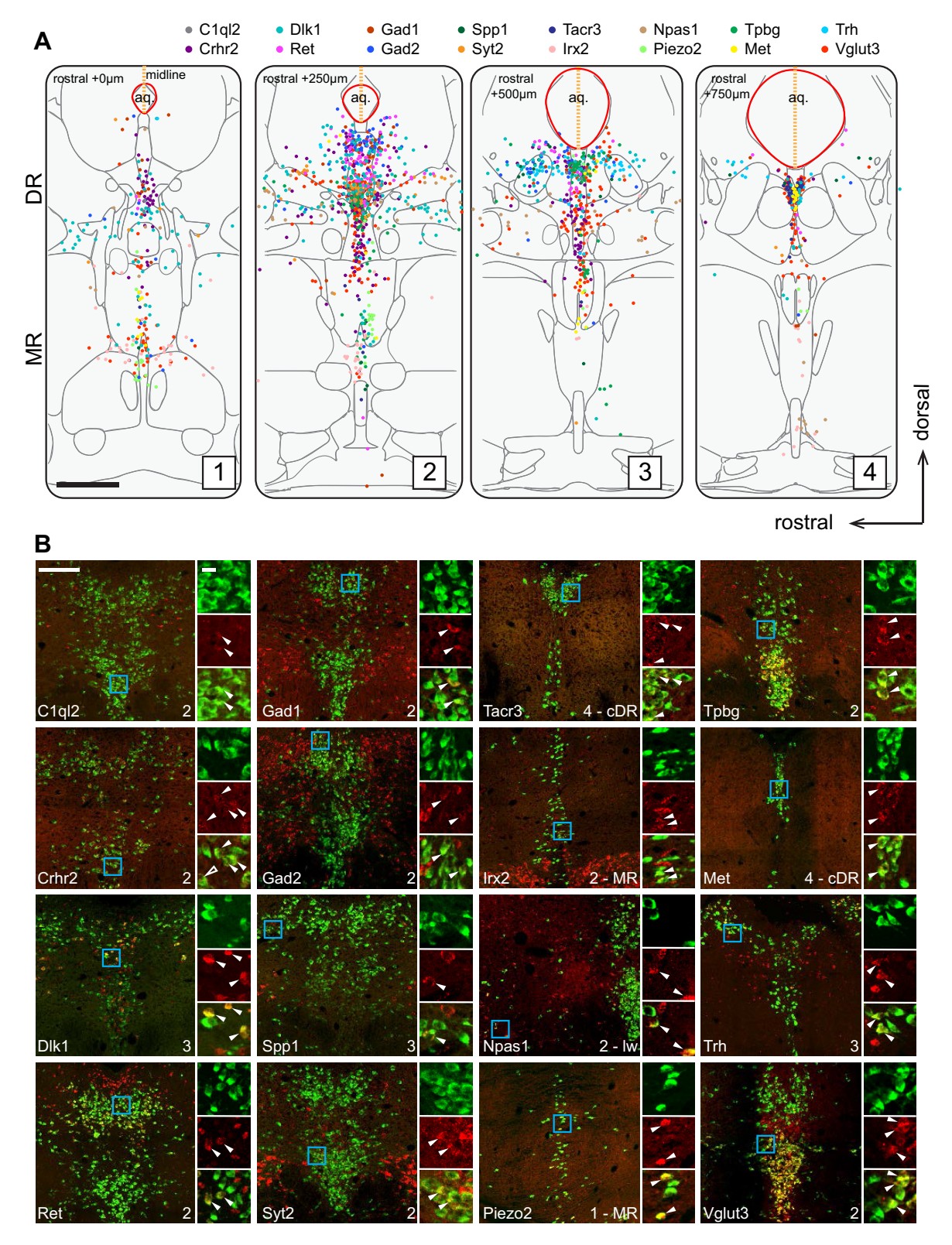

**Figure 2.** Anatomical location of serotonin neuron clusters determined by hybridization-chain reaction-based single-molecule fluorescence in situ hybridization (HCR-smFISH) of 16 cluster markers. (**A**) Positions of double-positive neurons (*Tph2* and each of 16 marker genes color-coded on the top) are shown on four schematics representing coronal slices 250 μm apart. Red line around the aqueduct represents the average boundary drawn from raw data for each slice. Scale bar, 500 μm. Schematics summarizing HCR-smFISH data for individual genes can be found in *Figure 2—figure*

*Figure 2 continued on next page*

*Figure 2 continued*

*supplements 1* and *2*. (B) Representative images for each of the genes schematized in (A). All green cells are Tph2-positive, red cells express the indicated marker gene. Each image corresponds to one of the four numbered schematics in (A) and is located immediately ventral to the aqueduct unless otherwise noted as MR, lateral wing (lw), or cDR. Cyan box highlights the individual color zoom region at right. White arrowheads mark examples of double-positive neurons. Scale bars, 200 µm main panels, 20 µm zoom panels.

DOI: https://doi.org/10.7554/eLife.49424.009

The following figure supplements are available for figure 2:

**Figure supplement 1.** Location of cluster marker genes determined by HCR-smFISH (Part I).

DOI: https://doi.org/10.7554/eLife.49424.010

**Figure supplement 2.** Location of cluster marker genes determined by HCR-smFISH (Part II).

DOI: https://doi.org/10.7554/eLife.49424.011

prefrontal cortex (*Ren et al., 2018*), nucleus accumbens (*Liu et al., 2014*), ventral tegmental area (*Wang et al., 2019*), and basolateral amygdala (*Sengupta et al., 2017*).

Three DR clusters express *Gad1* and *Gad2*, which encode biosynthetic enzymes for the neurotransmitter GABA. Two MR clusters express *Gad2* but not *Gad1*, and cDR expresses neither. Interestingly, few of the $Gad1^+$ or $Gad2^+$ neurons express vesicular GABA transporter (*Vgat*, *Figure 4A* left), which is responsible for transporting GABA into synaptic vesicles for synaptic transmission. It has been reported that vesicular monoamine transporters (*Slc8a1* for Vmat1; *Slc8a2* for Vmat2) can transport GABA into synaptic vesicles (*Stensrud et al., 2014*), and virtually all serotonin neurons expressed Vmat2 (*Figure 1—figure supplement 1D*; *Figure 4A* left). Nevertheless, it remains to be determined if these serotonin neurons can actually release GABA. At the single cell level, 14% of serotonin neurons do not express any of the gene markers for glutamate or GABA transmission. 65% of serotonin neurons express *Vglut3*, 35% express either *Gad1*, *Gad2*, or both, and 13% express markers for *Vglut3* and *Gad1* or *Gad2* (*Figure 4A* left).

In addition to small-molecule transmitters, the majority of serotonin neurons also co-express neuropeptides (*Figure 4A* right). Expression of genes encoding several neuropeptides served as excellent cluster markers. For example, *thyrotropin-releasing hormone* (*Trh*) is highly expressed in DR-1–3 (14% serotonin neurons, *Figure 1C*). *Corticotropin-releasing hormone* (*Crh*) is highly expressed in the DR-3 cluster but much less everywhere else (7% serotonin neurons). *Neuropeptide B* (*Npb*) is highly expressed in cDR and MR-4 but much less in pDR serotonin neurons. *Galanin* (*Gal*) is highly expressed in pDR and MR-1 (*Fernandez et al., 2016*). Many serotonin neurons express multiple neuropeptides (*Figure 4A,B*).

## Small-molecule neurotransmitter receptors

Each cluster has a distinct expression pattern of neurotransmitter (including neuropeptide) receptors (*Figure 4B*, *Figure 4—figure supplement 1*). Multiple subunits of glutamate and GABA receptors are differentially expressed across the clusters. For example, *Grin3a*, encoding subunit 3A of ionotropic NMDA glutamate receptor, is mainly expressed in pDR but not MR or cDR clusters. By contrast, *Grik1*, encoding subunit 1 of the kainate glutamate receptor, is more highly expressed in MR clusters. *Chrna3* and *Chrna4*, encoding subunit α3 and α4 of the nicotinic acetylcholine receptor, are also more enriched in MR clusters. Clusters MR-4 and cDR have the highest expression level of expression for *Chrm1* and *Chrm2*, encoding muscarinic acetylcholine receptors. *Dopamine receptor D2* (*Drd2*) is highly expressed in DR-3 (*Niederkofler et al., 2016*). Finally, all serotonin neurons express at least one type of serotonin receptors, with the $G_i$-coupled *Htr1* subfamily in particular *Htr1a* being the most widely expressed. *Htr1a* is highly expressed in the MR-4 cluster, which does not express other serotonin receptors. Of all the genes encoding serotonin receptors, *Htr5b* has the strongest cluster specificity (*Figure 4B*).

## Neuropeptide receptors

Multiple neuropeptide receptor-encoding genes have rich expression in distinct clusters. Notably, cDR preferentially expresses at least five neuropeptide receptors: *gastric inhibitory polypeptide receptor* (*Gipr*), *tachykinin receptor 3* (*Tacr3*), *relaxin family peptide receptor 1* (*Rxfp1*), *calcitonin receptor* (*Calcr*), and *opioid receptor mu-1* (*Oprm1*). Meanwhile, *opioid receptor kappa 1* (*Oprk1*) is

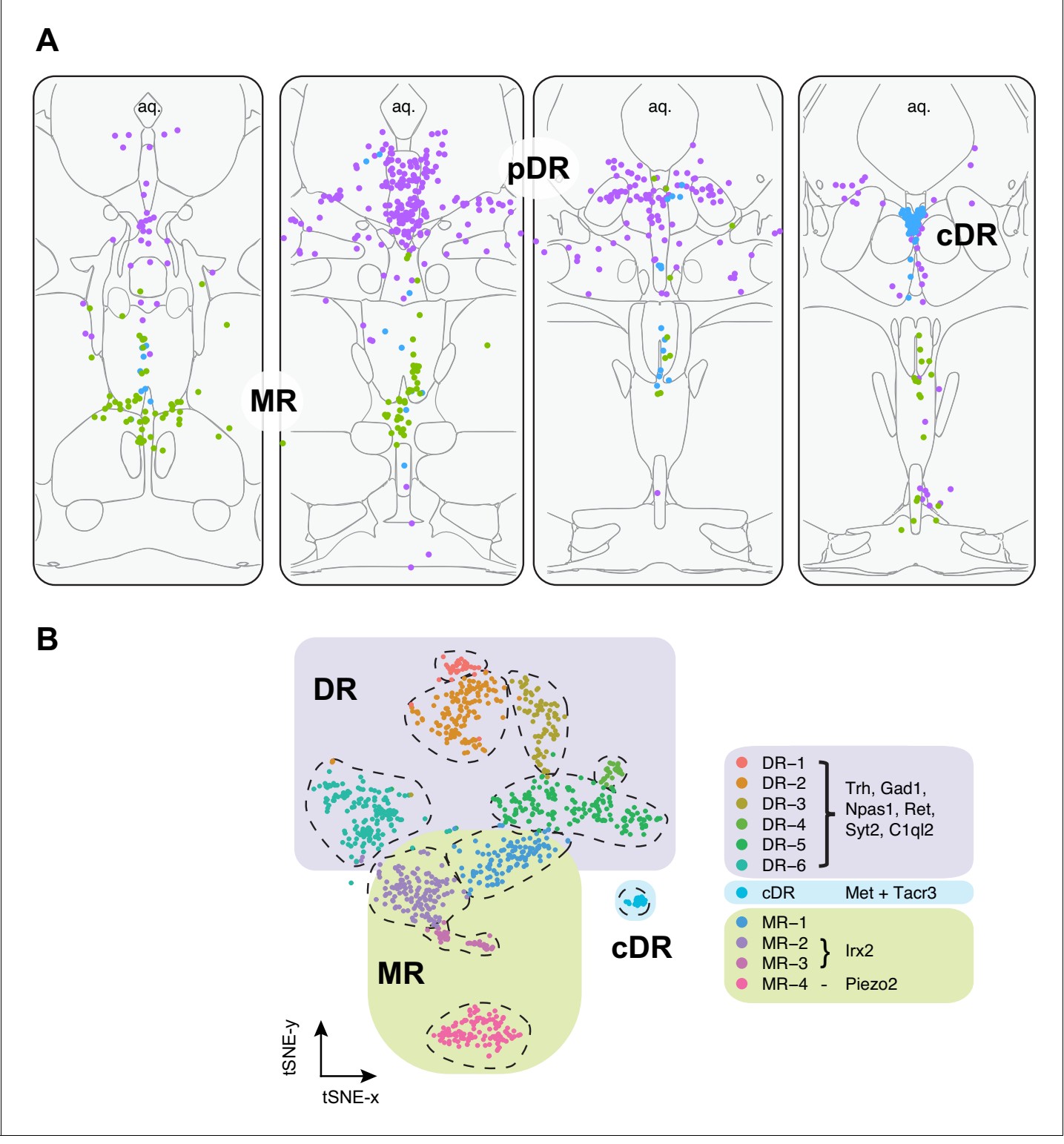

**Figure 3.** Summary of spatial distribution of transcriptomic clusters of serotonin neurons. (**A**) Purple dots represent distribution of Clusters 1–6 markers listed in panel (**B**); green dots represent the distribution *Irx2* and *Piezo2* cells that are Cluster 9–11 markers; and cyan dots present distribution of *Met*[+] and *Tacr3*[+] cells, which are co-expressed in Cluster 7, but is additionally expressed in Cluster 10 (*Met*) or Cluster 11 (*Tacr3*). (**B**) Collectively, scRNA-seq and HCR-smFISH experiments support the model that Clusters 1–6 from *Figure 1B* correspond to pDR serotonin neurons (renamed DR-1–6), Cluster 7 corresponds to cDR serotonin neurons, and Clusters 8–11 correspond to MR serotonin neurons (renamed MR-1–4).

DOI: https://doi.org/10.7554/eLife.49424.012

*Figure 3 continued on next page*

*Figure 3 continued*

The following figure supplements are available for figure 3:

**Figure supplement 1.** Expression of marker genes across 11 clusters (Part I).
DOI: https://doi.org/10.7554/eLife.49424.013
**Figure supplement 2.** Expression of marker genes across 11 clusters (Part II).
DOI: https://doi.org/10.7554/eLife.49424.014
**Figure supplement 3.** Expression of marker genes across 11 clusters (Part III).
DOI: https://doi.org/10.7554/eLife.49424.015
**Figure supplement 4.** Comparison of Marker Genes for Subtypes of MR Serotonin Neurons.
DOI: https://doi.org/10.7554/eLife.49424.016

expressed specifically in MR-2. Another neuropeptide receptor enriched cluster is MR-3, expressing *Tacr3*, *cannabinoid receptor 1*(*Cnr1*), *somatostatin receptor-1* (*Sstr1*), *prolactin-releasing hormone* receptor (*Prlhr*), and *neuromedin B receptor* (*Nmbr*). Neuropeptide receptors like *oxytocin receptor* (*Oxtr*), *prokineticin receptor-1 and −2* (*Prokr1*, *Prokr2*), *hypocretin receptor-2* (*Hcrtr2*), *corticotropin-releasing hormone receptor-2* (*Crhr2*), *cholecystokinin-A* (*Cckar*), *neuropeptide Y receptor Y2* (*Npy2r*), and *parathyroid hormone-2 receptor* (*Pth2r*) all have distinct expression pattern across different clusters. In summary, the DR and MR serotonin neurons have diverse expression pattern of neuropeptide receptors, indicating that they are subject to complex modulation by a host of neuropeptides.

## Other notable genes

The DR and MR serotonin clusters are also distinguished by differential expression of ion channels as well as axon guidance and cell adhesion molecules (*Figure 4—figure supplement 1*), which can contribute to their differences in physiological properties and wiring specificity. Notably, genes encoding a voltage-gated $K^+$ channel (*Kchn8*) and mechanosensitive channel (*Piezo2*) are highly expressed in the MR-4 cluster but exhibit little expression in all other clusters. The *chemokine ligand-12* (*Cxcl12*) and *cadherin-6* (*Cdh6*) are preferentially expressed in MR-1 and DR-4 clusters, respectively.

## Transcription factors

Transcription factors (TFs) have been shown to be the best molecular feature for cell type definition (*Schaum et al., 2018*). Within our data, we observed robust cluster-specific expression of multiple TF genes (*Figure 4B*; *Figure 4—figure supplement 1*; *Figure 3—figure supplement 1–3*). For example, *Bcl11a* and *Bcl11b* (also named *Ctip1* and *Ctip2*) (*Chen et al., 2008*; *Wiegreffe et al., 2015*) are highly enriched in DR-3 and MR-4, respectively. *Nfix* and *Nfib* (*Bedford et al., 1998*) are preferentially expressed in DR-3. *Irx2* (*Wylie et al., 2010*) is specific to MR-2 and MR-3. *Sox13* is highly enriched in cDR. Interestingly, *Sox14*, previously reported to be associated with GABAergic neurons in developing brain (*Lahti et al., 2016*), is expressed in Clusters 1 and 2 in adult serotonin neurons that co-express *Gad1* and *Gad2*. *Zeb2* (*Okaty et al., 2015*) is uniquely expressed in cDR and MR-4. TFs associated with neurodevelopmental disorders, such as *Npas1* and *Npas3* (*Erbel-Sieler et al., 2004*), are preferentially expressed in DR-6, and *Aff2* (*Mondal et al., 2012*) is enriched within DR-3.

## Transcriptional networks

To understand the relationship between cluster-specific genes and infer putative transcriptional regulatory programs orchestrating the maintenance of serotonin neuron subtype identity, we next performed a pairwise correlation analysis of gene expression across all 999 neurons (*Supplementary file 3*). We used Pearson correlation coefficient ($r_p$) as a measure of gene co-expression and focused on the genes with average expression >10 counts per million. We found multiple genes to co-expressed (defined as $r_p$ >0.5) in serotonin neurons, forming co-expression modules among each other as well as around various TFs (*Supplementary file 4*). Focusing further on cluster markers, we identified three transcriptional 'hubs' that putatively govern molecular programs within respective neurons. Interestingly, two of these hubs were centered around pDR-specific TFs, and, independently, comprised of dorsal (DR-1, DR-2, DR-3) and ventral (DR-4, DR-5, DR-6) pDR markers

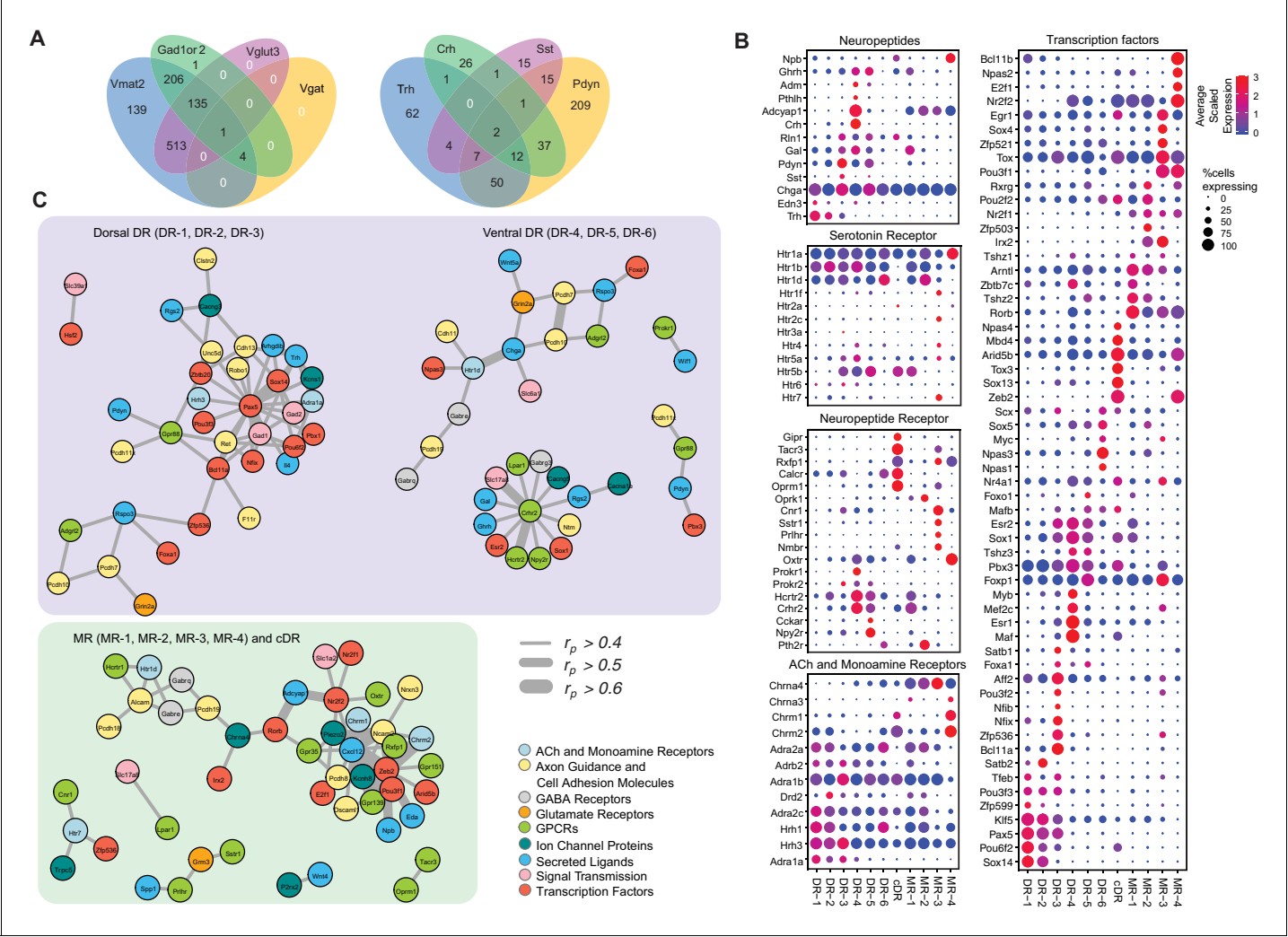

**Figure 4.** Molecular and functional characteristics of distinct serotonin neuron clusters. (**A**) Venn diagram showing the number of cells co-expressing genes associated with markers for different neurotransmitter systems: *Gad1/2*, *Vmat2*, *Vgat*, and *Vglut3* (left) and neuropeptides: *Trh*, *Crh*, *Sst*, *Pdyn* (right). We consider a gene to be expressed if it has at least one read mapping to it and is detected in at least 3 cells (Materials and methods). (**B**) Expression of the most variable neuropeptides, receptors, and transcription factors across molecularly distinct serotonin neuron clusters. (**C**) Network representation of co-expressed genes that belong to one of the functional gene categories that organize transcriptional regulation, synaptic connectivity, and neuronal communications. Networks were constructed based on Pearson correlation coefficient ($r_p$) of gene expression across all 999 cells, and identified networks that are centered on pDR- and MR+cDR-specific genes. Genes appear connected if $r_p$ >0.4. Line width represents $r_p$ values as indicated. Nodes are colored according to functional gene categories.

DOI: https://doi.org/10.7554/eLife.49424.017

The following figure supplements are available for figure 4:

**Figure supplement 1.** Expression of the most variable genes among the listed functional categories across 11 molecularly distinct *Tph2+* clusters.
DOI: https://doi.org/10.7554/eLife.49424.018

**Figure supplement 2.** Sexually dimorphic genes consistently detected across the majority of serotonin cell subtypes.
DOI: https://doi.org/10.7554/eLife.49424.019

(*Figure 4C*). Among dorsal pDR markers we identified *Pax5* and *Sox14* to strongly correlate ($r_p$ >0.5) with *Gad1*, *Gad2*, and *Trh*, among other genes. Among ventral pDR markers, we found *Crhr2* to be highly correlated with several TFs, receptors, and neurotransmitter-related genes, most notably *Vglut3* (*Slc17a8*) and *Hcrtr2*. These *Crhr2+* serotonin neurons could use TFs *Esr2* and *Sox1* to maintain their subtype identity. *Npas3* is specifically expressed in the DR-6 cluster and is highly correlated with *Htr1d*, suggesting a critical role of maintaining the characteristics of DR-6 serotonin neurons. Similarly, we found several MR- and cDR-specific TFs to be the hubs of co-expression modules.

Particularly, the expression of a large number of cell adhesion molecules, receptors, ion channel proteins and neuropeptides strongly correlate with the expression of TFs *Zeb2*, *Pou3f1*, *Irx2*, and *Zfp536*. Based on the identified correlation among gene expression of multiple marker genes across all the cells, we speculate that the identified genes could be linked by one or multiple transcriptional regulatory programs, potentially driving cell type-specific functions of distinct serotonin neuron populations.

## Sexually dimorphic gene expression

Finally, even though there were no apparent sex-specific difference at the cluster level *Figure 1—figure supplement 1D*), we did detect several genes, such as *Sod1*, *Snx10, Inpp4a, Zscan26, Ncam1,* that showed sexual dimorphism across the majority of cell subtypes (*Figure 4—figure supplement 2*).

## Viral-genetic tools to access different serotonin neuron subtypes

Gene expression patterns of specific serotonin neuron clusters can in principle allow genetic access to these specific subpopulations for anatomical tracing, physiological recording, and functional perturbation (*Luo et al., 2018*). However, DR and MR contain not only serotonin neurons but also GABAergic and glutamatergic neurons that do not express Tph2 (and hence do not release serotonin), some of which may project to the same target regions (*McDevitt et al., 2014*). To precisely investigate the function of transcriptome-based serotonin neuronal types, we need to use an intersectional strategy to target serotonin neurons that express specific additional markers (*Jensen et al., 2008*). To this end, we generated *Sert-Flp* mice through homologous recombination-based knock-in in embryonic stem cells (Materials and methods), and used *Sert-Flp* mice to intersect with transgenic Cre mice that allow expression of a fluorescent reporter only in Flp$^+$/Cre$^+$ (AND gate), so as to genetically label only specific Cre-positive clusters (*Figure 5A*).

To characterize the *Sert-Flp* mouse line, we crossed it with *H11-CAG-FRT-stop-FRT-EGFP* mice we newly generated (Materials and methods). Anti-Tph2 staining on the brain slices containing pDR showed that 98.5% GFP$^+$ neurons are Tph2$^+$ and 100% Tph2$^+$ neurons are GFP$^+$ (*Figure 5B*). To further verify the intersectional strategy and to label the serotonin neurons co-expressing markers for glutamate or GABA transmission, we crossed *Sert-Flp* with the *IS* reporter mice (*Rosa-CAG-loxP-stop-loxP-FRT-tdTomato-FRT-EGFP*) (*He et al., 2016*) and either *Vglut3-Cre* (*Grimes et al., 2011*) or *Gad2-Cre*. Anti-Tph2 staining showed that all GFP-labeled neurons are Tph2$^+$ (*Figure 5C,D*). In the pDR, Vglut3$^+$ serotonin neurons were mainly located ventrally, whereas Gad2$^+$ serotonin neurons were mainly located dorsally, consistent with our previous study (*Ren et al., 2018*) and the HCR-smFISH results (*Figure 2*).

To map the axonal projection pattern of serotonin subtypes defined by intersection of Flp and Cre expression, we developed a new AAV vector (*AAV-CreON/FlpON-mGFP*) that expressed membrane-targeted GFP under the dual gates of Flp and Cre (*Figure 6A*). Based on our scRNA-seq and HCR-smFISH results, the *Vglut3$^+$* and *Trh$^+$* pDR serotonin neurons consist of largely complementary cell types at the transcriptomic level and have a distinct distribution along the dorsal–ventral axis in the pDR. To visualize these two subpopulations of serotonin neurons, we injected *AAV-CreON/FlpON-mGFP* into pDR of either *Vglut3-Cre;Sert-Flp* (n = 3, *Figure 6B*) or *Trh-Cre;Sert-Flp* mice (n = 3, *Figure 6C*); *Krashes et al., 2014*. Anti-Tph2 staining showed that 98.2% GFP$^+$ neurons were Tph2$^+$. As predicted, *Vglut3$^+$Sert$^+$* GFP cells were mostly located in the ventral pDR (*Figure 6B*), whereas *Trh$^+$Sert$^+$* cells were located in the dorsal pDR (*Figure 6C*). As negative controls, we injected the same virus into mice carrying only the *Sert-Flp* transgene or only the *Vglut3-Cre* transgene and did not find any mGFP$^+$ cell bodies or fibers (n = 3 for each; data not shown).

The intersectional strategy allowed us to trace the projection of GFP$^+$ axons from these two groups of serotonin neurons across the brain. We next examined their projections by staining 50 µm coronal sections every 200 µm across the brain with anti-GFP antibody. We found that serotonin axons from *Vglut3$^+$* population preferentially targeted cortical regions (*Figure 6D*), consistent with our previous results (*Ren et al., 2018*). By contrast, no GFP-labeled axons were observed in the cortical regions from *Trh-Cre;Sert-Flp* mice. Instead, *Trh$^+$* serotonin axons project to the anterior and medial hypothalamus, posterior amygdala, and the lateral geniculate nucleus in the thalamus, none of which were targeted by *Vglut3$^+$* axons (*Figure 6D*).

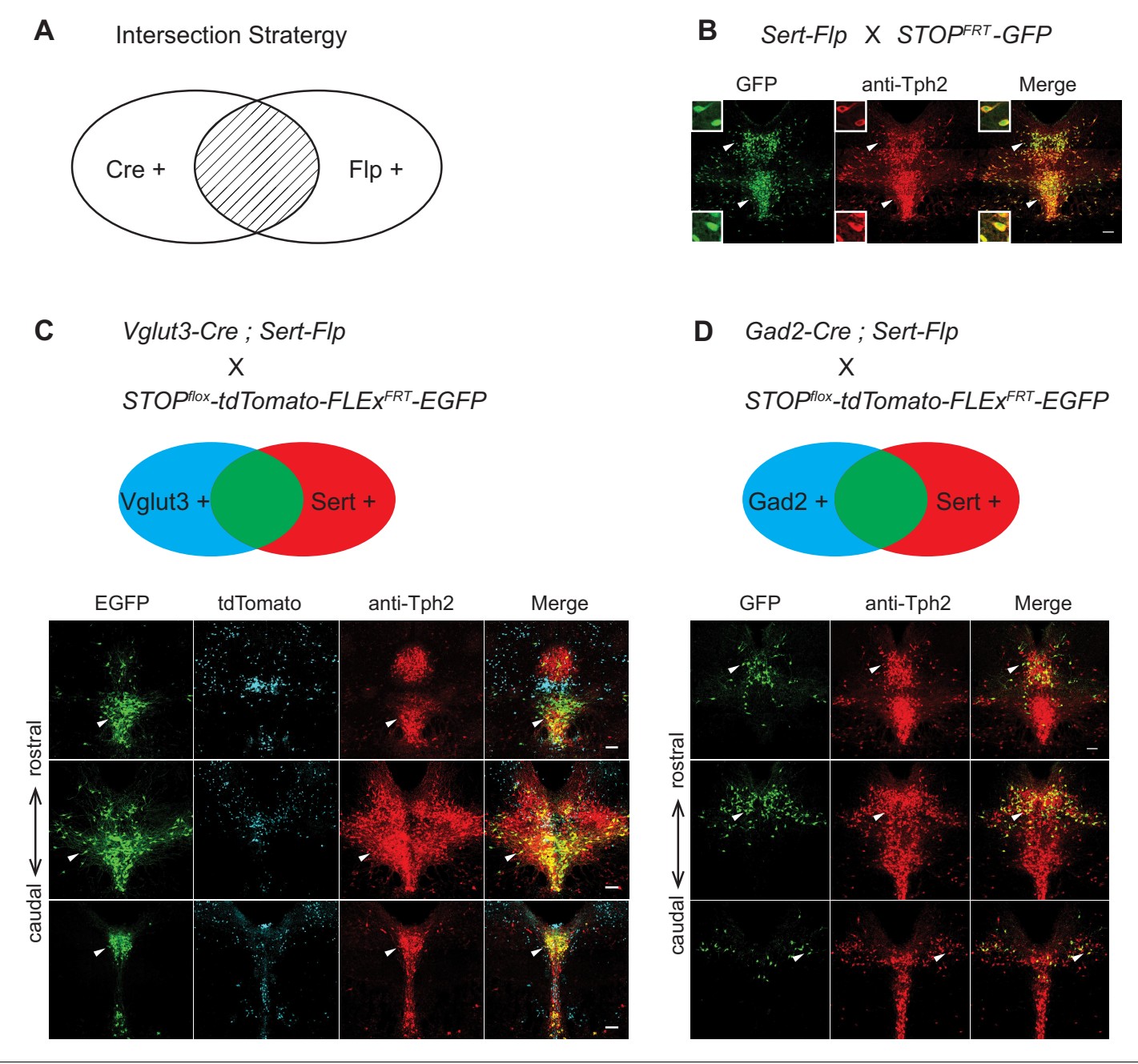

**Figure 5.** Intersectional strategy to genetically access specific serotonin neuron populations. (A) Schematic representing cells dually gated by Cre- and Flp-expression. (B) *Sert-Flp* mice were crossed with *H11-CAG-FRT-STOP-FRT-EGFP* (*STOP^FRT-GFP*) mice. Anti-Tph2 staining (red) was performed on consecutive coronal sections containing DR. 98.5% GFP$^+$ neurons are Tph2$^+$ and 100% Tph2$^+$ neurons are GFP$^+$ (n = 3 mice). Insets: magnified images showing the neurons indicated with arrowheads in individual channels. Scale, 25 µm. (C) In mice triple transgenic for *Vglut3-Cre*, *Sert-Flp*, and the *IS* reporter (*Rosa-CAG-loxP-stop-loxP-FRT-tdTomato-FRT-EGFP*), EGFP$^+$ (Flp$^+$Cre$^+$) cells are referentially found in ventral pDR and in cDR (arrowheads). Coronal sections containing DR are shown, counterstained with Anti-Tph2 (red). (D) In mice triple transgenic for *Gad2-Cre*, *Sert-Flp*, and the *IS* reporter, EGFP$^+$ (Flp$^+$Cre$^+$) cells are referentially found in dorsal pDR (arrows). Coronal sections containing DR are shown, counterstained with Anti-Tph2 (red). Scale, 100 µm.

DOI: https://doi.org/10.7554/eLife.49424.020

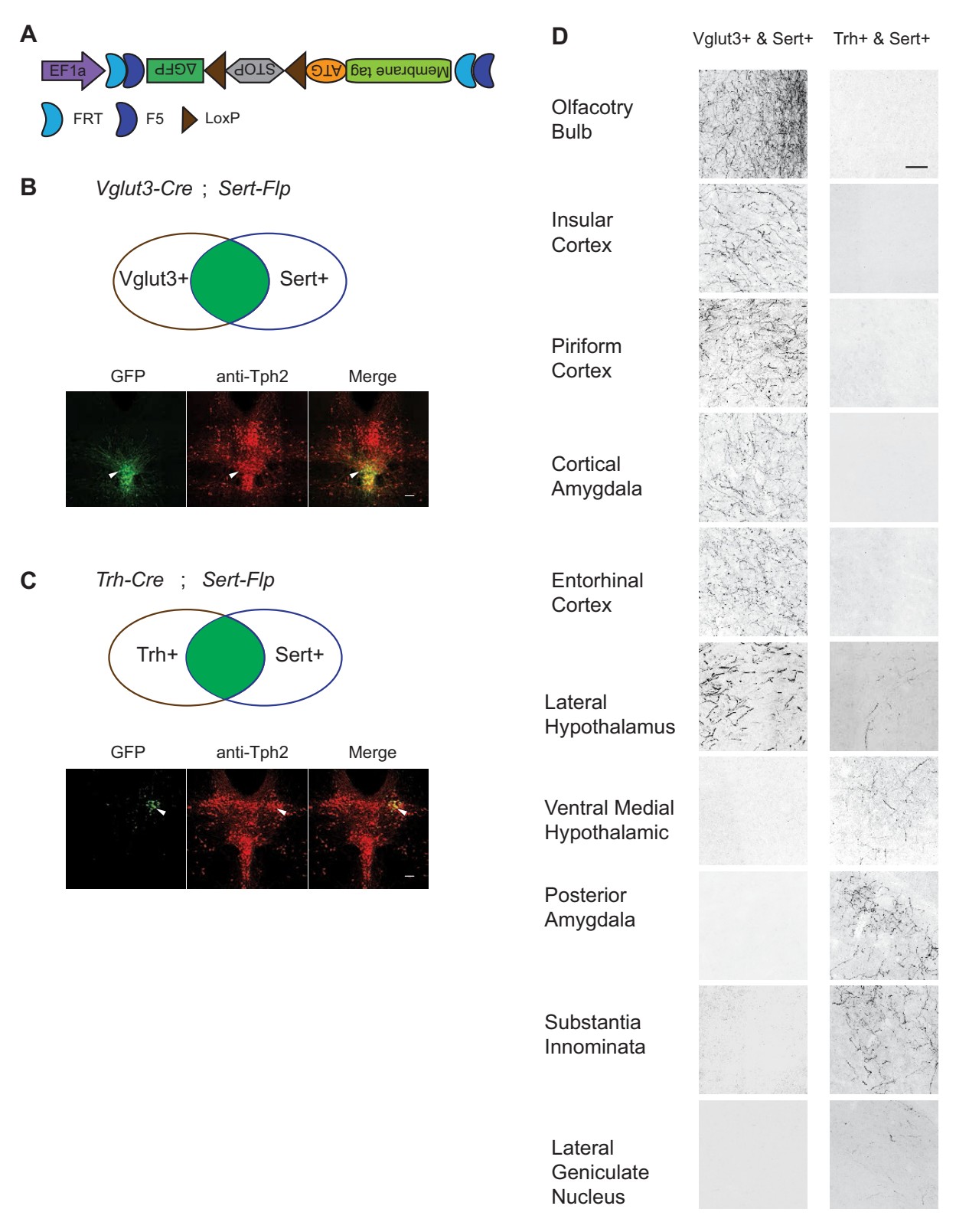

**Figure 6.** Dually gated serotonin neuron and axonal terminal labeling by viral-genetic intersection. (**A**) Schematics representing of the *AAV-CreON/ FlpON-mGFP* dual labeling design. (**B**) mGFP expression (green) and Tph2 immunoreactivity (red) after injection of *AAV-CreON/FlpON-mGFP* into the DR of *Sert-Flp;Vglut3-Cre* mice. mGFP is mostly restricted to ventral pDR. (**C**) mGFP expression (green) and Tph2 immunoreactivity (red) after injection of *AAV-CreON/FlpON-mGFP* into the DR of *Sert-Flp;Trh-Cre* mice. mGFP is mostly restricted to dorsal pDR; the left-right asymmetry was likely due to

*Figure 6 continued on next page*

*Figure 6 continued*

AAV injection being biased towards the right hemisphere. (D) Axonal terminal expression of mGFP in different brain regions of *Sert-Flp;Vglut3-Cre* mice and *Sert-Flp;Trh-Cre* mice. The left and right images represent comparable brain regions cropped from serial coronal sections. Scale, 100 µm.

DOI: https://doi.org/10.7554/eLife.49424.021

## Whole-brain axonal projections of selected serotonin neuron subpopulations

While suggestive of an anatomical division in targets, assessing the full extent to which projections of *Vglut3+* or *Trh+* pDR serotonin neuron populations segregate requires quantifying axonal innervation at the whole-brain level. We used the iDISCO-based brain clearing technique AdipoClear (*Chi et al., 2018*) to visualize, align, and summarize whole-brain projectomes (*Figure 7A*). Individual hemispheres of either *Vglut3-Cre;Sert-Flp* (n = 3) or *Trh-Cre;Sert-Flp* mice (n = 3) injected with *AAV-CreON/FlpON-mGFP* at pDR were imaged by light-sheet microscopy. We developed deep learning models to automatically trace whole-brain axonal projections by segmenting volumes with a 3D U-Net-based convolutional neural network we developed (Materials and methods). The resulting volumetric probability maps were thinned and thresholded before aligning to the Allen Institute's 2017 common coordinate framework as previously described (*Ren et al., 2018*; *Figure 7A*).

We visualized the axon terminals in brain regions targeted by either the *Trh+* or the *Vglut3+* population of serotonin neurons. Initial assessment of selected target regions suggested a strong segregation of axonal projection patterns between these two populations (*Figure 7B,C*), consistent with data from tissue sections (*Figure 6D*). Axons from both populations followed similar initial trajectories and were observed in shared structures along the length of the median forebrain bundle (*Figure 7C,D*; *Figure 7—figure supplements 1–3*; *Figure 7—Video 1*). However, as was seen in tissue sections, there were extensive differences in innervation patterns of terminal axon fields between the *Vglut3+* and *Trh+* populations. Whole-brain quantitative and statistical analyses showed *Vglut3+* axons preferentially in anterolateral cortical regions and adjacent structures such as olfactory bulb, agranular insular cortex, endopiriform, piriform, and claustrum as well as other cortical regions such as entorhinal, primary motor, and barrel cortices. [Note that axons seen in hippocampus likely originated from *Vglut3+* cells in MR (*Muzerelle et al., 2016*), though the majority of labeled somata were located in the ventral portion of pDR; see *Figure 7—figure supplement 3C*]. By contrast, Trh+ axons, which originated preferentially from the lateral wing of pDR, were largely absent from these structures. Conversely, subcortical regions primarily in thalamus (zona incerta and medial geniculate) and hypothalamus (anterior and dorsomedial nuclei) were preferentially targeted by *Trh+* axons and largely avoided by the *Vglut3+* population (*Figure 7E*; *Figure 7—Video 1*).

Given the variability of locations and amount of viral transduction, individual brains from the same genotype exhibit considerable variation in total axons labeled (*Figure 7E* top) and in detailed projection patterns (*Figure 7—figure supplements 1–2*). These variabilities further highlighted regions that were targeted densely but exclusively in one or two individual brains. Notable examples include anterior bed nucleus of stria terminalis (BNST), posterior amygdala, and globus pallidus external segment (GPe) in *Trh+* projections, as well as the lateral central amygdala (CeA) and dentate nucleus of the cerebellum for *Vglut3+* projections. While we did identify large-scale patterns of collateralization for these two subtypes of serotonin neurons, one possible contribution to this inter-individual variability in projection patterns is heterogeneity within molecularly defined subpopulations of serotonin neurons. Larger-scale experiments or experiments using Cre lines that are expressed in single clusters will be required to further dissect heterogeneity within the *Vglut3+* or *Trh+* serotonin neurons.

## Whole-brain axonal arborization patterns of individual serotonin neurons

Our whole-brain projection analyses indicate that axonal arborization patterns of molecularly defined serotonin neuron subpopulations are still very complex (*Figure 7*). To examine the extent to which this reflects projection patterns of individual serotonin neurons, we combined the cell-type-specific sparse labeling strategy we recently developed (*Lin et al., 2018*) (*Figure 8—figure supplement 1A*) and the fluorescence micro-optical sectioning tomography (fMOST) platform (*Gong et al., 2016*). We fully reconstructed whole-brain arborization patterns of 50 DR serotonin neurons from 14 *Sert-*

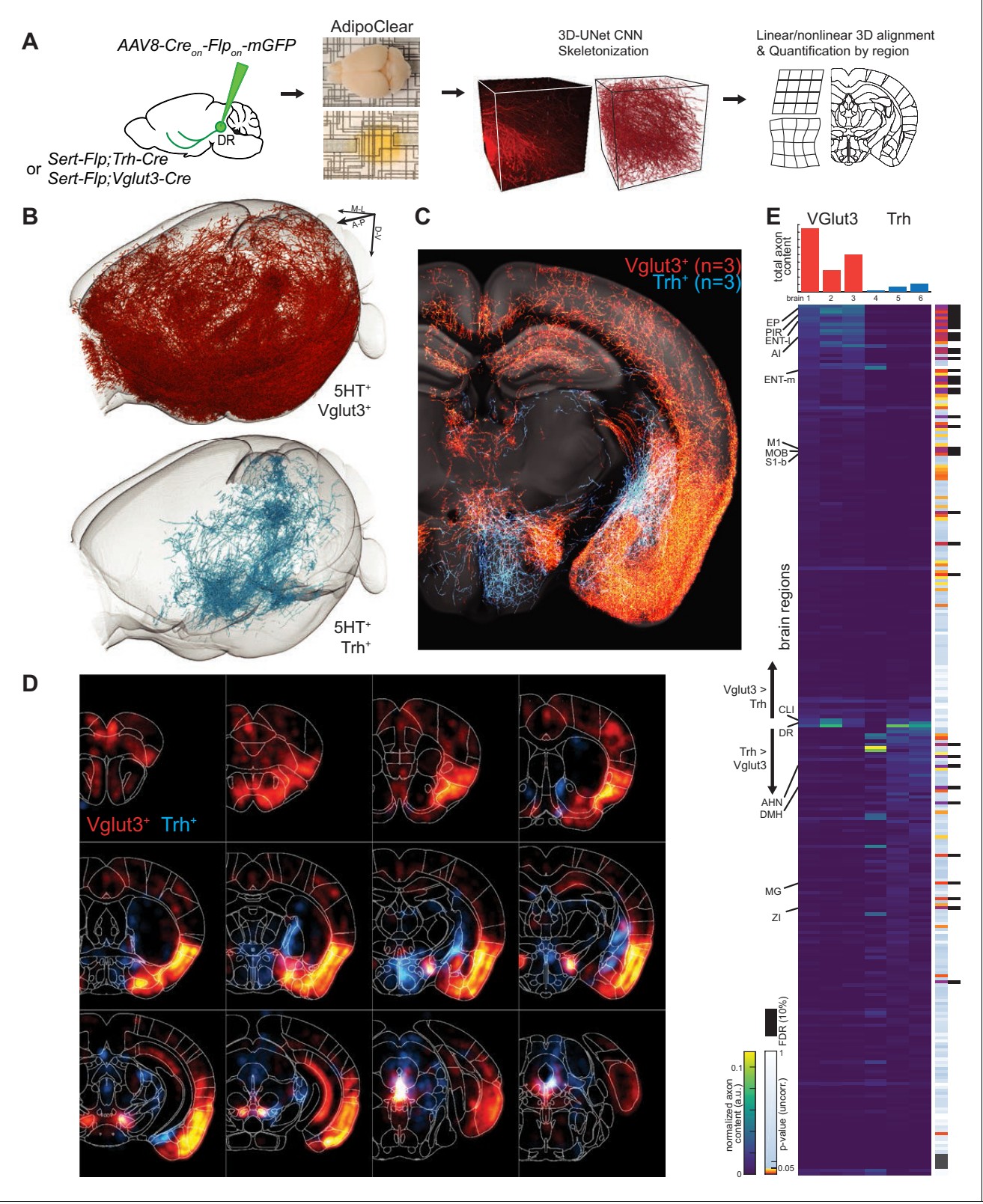

**Figure 7.** Whole-brain projectome of *Trh+* and *Vglut3+* serotonin neuron populations. (**A**) Experimental schematic outlining the intersectional viral strategy, brain clearing, automated 3D axon segmentation, and alignment to the Allen Brain Institute Common Coordinate Framework. (**B**) Axonal innervation in a 3D view of the left hemisphere of one representative brain each from the intersection of *Sert-Flp* and either *Vglut3-Cre* or *Trh-Cre*. (**C**) Coronal Z-projection (500 μm of depth) showing axonal innervation patterns of 6 aligned brains. The schematic reference image is one 5 μm thick plane

*Figure 7 continued on next page*

Figure 7 continued
in the middle of the 500 μm stack. (D) Coronal heatmaps of axonal innervation patterns at 12 positions along the rostral–caudal axis for the same six brains as seen in (C). Weightings for individual voxels represent axonal content within a radius of 225 μm. (E) Top, bar plot shows the quantification of total axonal content in each of 6 brains prior to normalization. Bottom, heatmap breaks out the total content into each of 282 individual brain regions using boundaries from the Allen Institute CCF. Values are normalized to both target region volume and total axon content per brain. Display order is grouped by mean normalized prevalence of axons in each genotype and ordered by the second principal component. P-values for individual t-tests are uncorrected; those that survive FDR-testing at 10% are indicated with a black bar. See **Supplementary file 5** for full list of regions. EP, Endopiriform nucleus; M1, Primary motor area; S1-B, Primary somatosensory area, barrel field; CLI, Central linear nucleus raphe; AHN, Anterior hypothalamic nucleus; DMH, Dorsomedial nucleus of the hypothalamus; MG, Medial geniculate nucleus; ZI, Zona incerta.

DOI: https://doi.org/10.7554/eLife.49424.022

The following video and figure supplements are available for figure 7:

**Figure supplement 1.** Individual projectome variability of the Trh[+] serotonin population.

DOI: https://doi.org/10.7554/eLife.49424.023

**Figure supplement 2.** Individual projectome variability of the Vglut3[+] serotonin population.

DOI: https://doi.org/10.7554/eLife.49424.024

**Figure supplement 3.** The cell body location and initial axonal segments of the *Trh[+]* and *Vglut3[+]* serotonin neuron subpopulations.

DOI: https://doi.org/10.7554/eLife.49424.025

**Figure 7—video 1.** Fly-through of aligned axonal projections and heatmaps of the *Trh[+]* and *Vglut3[+]* serotonin subpopulations.

DOI: https://doi.org/10.7554/eLife.49424.026

*Cre* mice. All reconstructed neurons and their processes were registered to the Allen Reference Atlas (**Figure 8A**) (**Gilbert, 2018**). The cell bodies of these 50 serotonin neurons covered large regions of the DR (**Figure 8—figure supplement 1B**), and their projections collectively innervated the majority of the brain regions (**Figure 8—figure supplement 1C**), suggesting their broad coverage of DR serotonin neuron types. Complete morphological reconstruction revealed that the projection pattern of individual DR serotonin neurons was highly diverse yet follow some general patterns (**Figure 8—figure supplement 1C**). For the purpose of discussion below, we categorized them into six groups based on their projection patterns. The groups were named after the brain region innervated by the highest proportion of their terminals, even though many neurons project to more than one brain region categorized below (**Figure 8B–G**; **Figure 8—figure supplements 2–3**; **Figure 8— Video 1**; Materials and methods).

## Cortex-projecting
Cortical regions contain the highest proportion of the terminals of this group of DR serotonin neurons (n = 17), which is more than 40% of their total axonal length. In general, this group had the most complex branching pattern, with various combinations of target regions (**Figure 8B,G**; **Figure 8—figure supplement 2A**). This group could be further divided into four subgroups based on the dominant targets (containing >40% total axon length in cortical regions): olfactory area-projecting (including olfactory bulb, piriform cortex, and cortical amygdala; n = 9), prefrontal cortex-projecting (n = 2), dorsal cortex-projecting (including motor, somatosensory, retrosplenial, auditory, and visual cortex; n = 4), and ventrolateral cortex-projecting (including entorhinal, perirhinal, and ectorhinal cortex; n = 2). In the olfactory area-projecting subgroup, three brains had branches in the olfactory bulb (OB). Axons of the three dorsal cortex-projecting neurons traveled the largest distance, entering the cortex anteriorly and extending to the posterior end. Interestingly, two cortex-projecting serotonin neurons also sent substantial branches to the pons (>10% of axon length), and also innervated the cerebellum.

## Striatum-projecting
Each cell of this group (n = 5) dedicated more axon terminals to the striatum, pallidum, and lateral septal nucleus (>20% of the axon length) than other brain regions except midbrain regions (**Figure 8B**; **Figure 8—figure supplement 2B**). One cell had substantial branches to the cortical regions and two to the hypothalamus. One cell had extensive collateralization in the central amygdala (CeA) (>10% axon length) (**Figure 8—figure supplement 4**). For two DR serotonin neurons that innervated ventral striatum (also called nucleus accumbens, or NAc), arborization appeared to be restricted to either the core or the shell (**Figure 8—figure supplement 4C**).

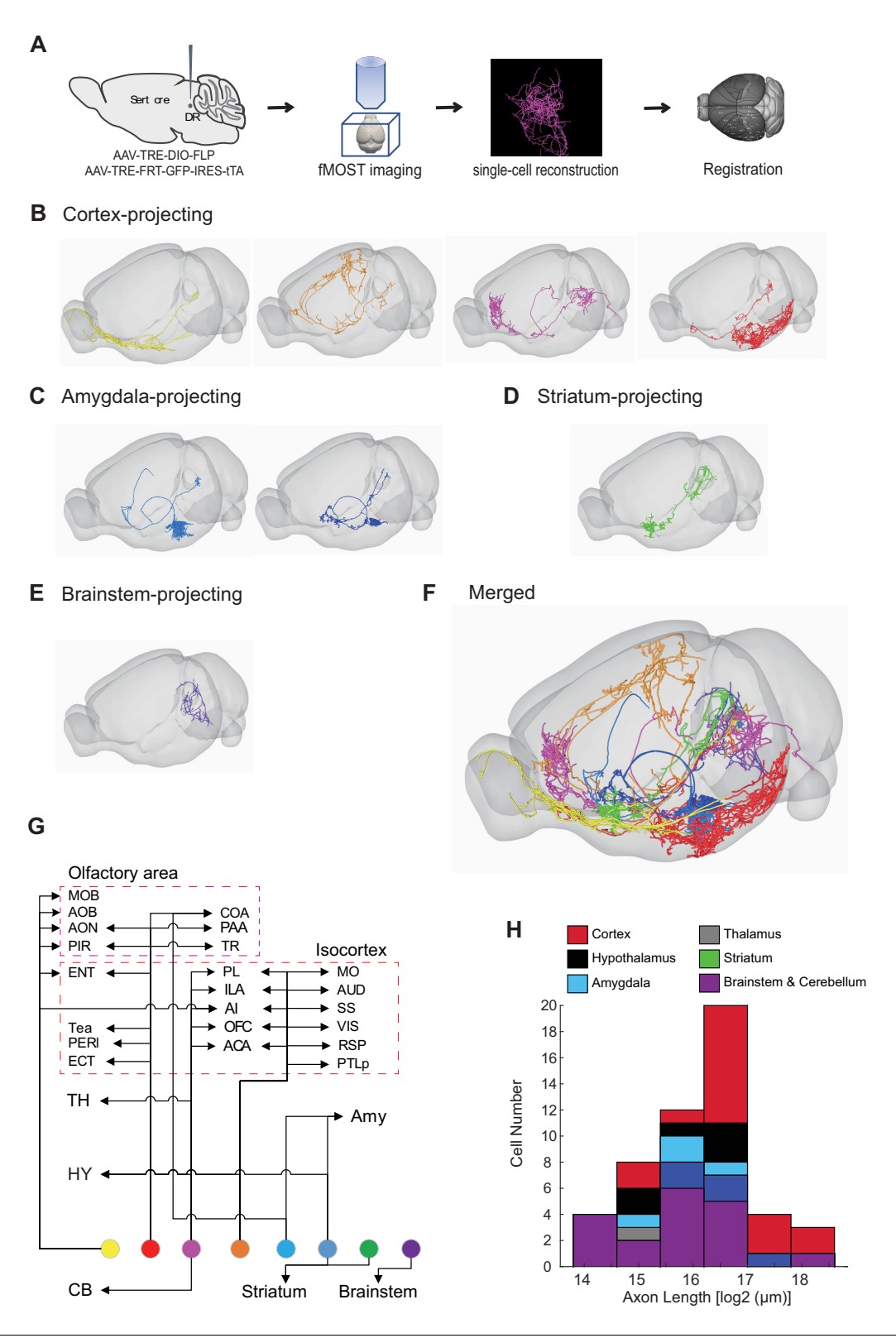

**Figure 8.** Whole-brain axonal arborization patterns of individual serotonin neurons. (**A**) Schematic of single-neuron reconstruction pipeline. (**B**) Four examples of cortex-projecting DR serotonin neurons, projecting primarily to olfactory cortex and olfactory bulb (1st), dorsal cortex (2nd), prefrontal cortex (3rd), and entorhinal cortex (4th). (**C**) Two examples of amygdala-projecting DR serotonin neurons. (**D**) A striatum-projecting DR serotonin neuron.
*Figure 8 continued on next page*

*Figure 8 continued*

(**E**) A brainstem-projecting DR serotonin neuron. (**F**) Merged example neurons from panels B–E. (**G**) Schematic diagram illustrating the major projection targets of 8 sample neurons in panel F. (**H**) Histogram showing the distribution of cell numbers according to the total axon length.

DOI: https://doi.org/10.7554/eLife.49424.027

The following video and figure supplements are available for figure 8:

**Figure supplement 1.** Single-cell reconstruction of DR serotonin neurons.

DOI: https://doi.org/10.7554/eLife.49424.028

**Figure supplement 2.** Individually reconstructed DR serotonin neurons (part I).

DOI: https://doi.org/10.7554/eLife.49424.029

**Figure supplement 3.** Individual reconstructed DR serotonin neurons (part II).

DOI: https://doi.org/10.7554/eLife.49424.030

**Figure supplement 4.** Detailed projection patterns of example DR serotonin neurons.

DOI: https://doi.org/10.7554/eLife.49424.031

**Figure 8—video 1.** Whole-brain axonal projection patterns of 6 reconstructed dorsal raphe serotonin neurons.

DOI: https://doi.org/10.7554/eLife.49424.032

### Hypothalamus-projecting

Each cell of this group (n = 6) dedicated more than 20% axon length to innervate the hypothalamus (*Figure 8B*; *Figure 8—figure supplement 2C*). None of them had collateralization in the cortical regions. Two cells had substantial branches to the striatum, and one to the pons (>10% axon length).

### Amygdala-projecting

This group of DR serotonin neurons (n = 4) all dedicated more than 40% their total axonal length to innervate the amygdala (*Figure 8C*; *Figure 8—figure supplement 3A*). None of them had collateralization in the cortical regions. Three cells had collateralized axonal terminals in the striatum, pallidum, and lateral septal nucleus. One cell had axons that were almost equally distributed in the bed nucleus of stria terminalis (BNST), basal lateral amygdala, basal medial amygdala, and medial amygdala (MeA), and had substantial branches to the hypothalamus (>10% axon length).

### Other groups

The rest of the DR serotonin neurons were divided into thalamus-projecting and brainstem/cerebellum-projecting groups based on the highest density of axons (*Figure 8—figure supplement 3B,C*), one of which likely projecting to the spinal cord (brain 36) as previously reported (*Bowker et al., 1981*; *Bowker et al., 1982*). While most projections of the DR serotonin neurons were unilateral (37/50), one of the thalamus-projecting neurons had symmetrical bilateral projections in the thalamic target (*Figure 8—figure supplement 4B*).

In summary, these results revealed remarkable complexity and heterogeneity of serotonin neuron projections at the single-cell level. They nevertheless followed certain patterns. For example, the cell body locations of these traced neurons (*Figure 8—figure supplement 1B*) were consistent with our previous results (*Ren et al., 2018*) as well as single-cell transcriptome, HCR-smFISH, and whole-brain bulk tracing data. Specifically, cortex-projecting serotonin neurons were biased towards ventral pDR (14 out of 17 cells), whereas amygdala-projecting ones were preferentially distributed in the dorsal pDR (3 out of 4 cells).

Forebrain-projecting serotonin neurons have been estimated to have very long axons based on the total serotonin fiber density and number of serotonin neurons (*Wu et al., 2014*). With single-cell tracing, we could now directly quantify the total length of axons of the 50 DR serotonin neurons. We found that the total axon length of these DR serotonin neurons exhibited considerable heterogeneity, from 1.9 cm to 26.3 cm. When examined across the six groups, cortex-projecting serotonin neurons indeed had the longest axons (*Figure 8H*). The longest axon from a single serotonin neuron (26.3 cm) is >10 times the length of a mouse brain in the longest dimension from rostral olfactory bulb to caudal brainstem.

## Discussion

Collectively,~12,000 serotonin neurons from the DR and MR in mice (~0.015% of all CNS neurons) innervate the entire forebrain (*Ishimura et al., 1988*) and modulate diverse physiological and behavioral functions. A fundamental question is how these serotonin neurons are organized to manage such a broad range of modulation. Single-cell transcriptomics have emerged in recent years as a powerful means to classify neuronal types, supplementing traditional criteria based on developmental history, morphology, projection patterns, and physiological properties (*Luo et al., 2018*). Using high-depth single-cell RNAseq in combination with systematic fluorescence in situ hybridization, whole brain projection mapping via intersectional methods and single-axon tracing, our study begins to shed light on the relationship between transcriptomic clusters, the spatial location of their cell bodies, and brain-wide projection patterns of serotonin neurons.

### Relationship between molecular identity and cell body distribution

Our single-cell transcriptome analysis identified 11 molecularly distinct types of serotonin neurons in the DR and MR (*Figure 1*). Based on tissue source from which scRNA-seq data was collected and fluorescent in situ hybridization using transcriptomic cluster markers, we were able to assign six types to principal DR (pDR), one type to caudal DR (cDR), and four types to MR (*Figure 3*). The fact that we can assign specific transcriptomic clusters to specific groups of raphe nuclei indicate that molecularly defined serotonin populations are spatially segregated at least at this coarse level.

The six types of serotonin neurons within the pDR exhibit further specificity in spatial distributions. Specifically, serotonin neurons from DR-1–3 clusters are preferentially localized in dorsal pDR, whereas those from DR-4–6 in ventral pDR, with DR-6 neurons preferentially localized to the ventral lateral wings. These data support and extend our previous finding (*Ren et al., 2018*) for the preferential ventral pDR location of $Vglut3^+$ serotonin neurons, which is highly expressed in DR-4 and DR-5 clusters. Together, these findings revealed the molecular basis for the differentiation of dorsal/ventral DR sub-systems (*Figures 2* and *3*).

The cDR has been suggested to be more similar to the MR than to the pDR in afferent innervation (*Commons, 2015*). By searching Allen Brain Institute development brain database and using multiplex fluorescent in situ hybridization (ML-FISH), *Kast et al. (2017)* found that the cDR expressed distinct gene markers comparing to pDR, and tracing from these neurons showed cDR was different from the MR and DR in efferent innervation. Our single-cell transcriptomic analysis indicated that serotonin neurons in the cDR are strikingly homogenous and profoundly different from both the pDR and MR at the molecular level (*Figure 1*, *Figure 4*, *Figure 4—figure supplement 1*). All cDR serotonin neurons co-express *Vglut3*, and express several unique markers (*Figure 3—figure supplement 2*). By contrast, we could not discern obvious difference in spatial distributions among the four MR types, despite the fact that MR serotonin neurons have heterogenous developmental origin (*Jensen et al., 2008*). It will be interesting to examine in the future whether molecularly defined MR serotonin neurons have specific axonal projection patterns.

Our results are broadly consistent with previous findings that utilized developmental origin to differentiate raphe serotonin neurons (*Okaty et al., 2015*). Okaty et al. used intersectional genetic fate mapping to produce population RNA-seq data (and single-cell RNA-seq data from a small number of cells) from DR, MR, and caudal raphe nuclei derived from different rhombomeres during development. Because DR serotonin neurons derive exclusively from rhombomere 1 (R1), and the single-cell RNA-seq data was obtained from just 8 cells, the Okaty et al. dataset is limited in providing information about heterogeneity within DR serotonin neurons. (Nevertheless, we confirmed the in situ hybridization data of Okaty et al. that *Met* is expressed in the cDR and MR.) The MR serotonin neurons derive from three separate rhombomeres (R1–R3), and Okaty et al. sampled these three populations separately, allowing us to ask the question of whether developmental origins correspond to transcriptomic clusters in adult. Comparing marker genes enriched in R1–R3-derived serotonin neurons from Okaty et al. and MR-1–MR-4 transcriptomic clusters from our current study (*Figure 3—figure supplement 4*) indeed suggests an approximate correspondence: R1-derived serotonin neurons correspond to our MR-1 cluster, R2 to MR-4 cluster, and R3 to both MR-2 and MR-3 clusters.

We note that in a recent comprehensive whole-brain scRNA-seq study, *Zeisel et al. (2018)* grouped 437 brainstem serotonin neurons in the dataset (presumably containing the entire B1–B9 groups) into just five clusters. While our manuscript in review, a related study was published

(*Huang et al., 2019*), identifying fewer transcriptomic clusters of DR serotonin neurons than we did. Compared to droplet-based platform of the two studies, our FACS-based scRNA-seq platform allowed more sequencing depth per cell (*Figure 1—figure supplement 1B*) and sequencing larger number of cells, accounting for our finer classification of transcriptomic types of serotonin neurons. This comparison illustrates the value of using genetically targeted strategies to characterize important but rare types of cells in the brain.

## Relationship between molecular identity and projection-defined serotonin sub-systems

In our previous study, we characterized two projection-defined parallel DR serotonin sub-systems. We found that serotonin neurons that project to the orbitofrontal cortex (OFC) and central amygdala (CeA) differ in input and output patterns, physiological response properties, and behavioral functions (*Ren et al., 2018*). Whole-brain collateralization patterns of these two sub-systems indicate that there must be additional sub-systems projecting to regions not visited by either of these two sub-systems projects to, including much of the thalamus and hypothalamus. What is the relationship between molecularly defined serotonin neurons and projection-defined sub-systems?

Using viral-genetic intersectional approaches to access specifically *Vglut3*⁺ pDR serotonin neurons in combination with staining in histological sections (*Figure 6*) and iDISCO-based whole brain imaging (*Figure 7*), we found that these neurons project profusely to cover much of the neocortex, as well as the olfactory bulb, cortical amygdala, and lateral hypothalamus. Comparisons of the projection patterns of *Vglut3*⁺ (this study) with OFC-projecting DR serotonin neurons (*Ren et al., 2018*) suggest that the latter is a large subset of the former. Brain regions that are innervated by *Vglut3*⁺ but not by OFC-projecting serotonin neurons include the somatosensory barrel cortex, ventral striatum, and a specific sub-region of CeA. The last finding is consistent with our previous study indicating that ~23% of CeA-projecting DR serotonin neurons are *Vglut3*⁺ (*Ren et al., 2018*).

We also assessed the whole-brain projection patterns of a largely complementary population of DR serotonin neurons, namely those that express *Trh* and thus belong to DR-1–3 clusters. We found that *Trh*⁺ serotonin neurons predominantly project to subcortical regions, most notably anterior and medial nuclei of the hypothalamus and several thalamic nuclei, a pattern mostly complementary to the *Vglut3*⁺ population (*Figures 6–7*, *Figure 7—Video 1*). Given that CeA-projecting DR serotonin neurons do not innervate most of the hypothalamus (*Ren et al., 2018*), and *Trh*⁺ serotonin neurons only partially innervate CeA, these two populations are at most partially overlapping.

These comparisons support a broad correspondence between molecular identity and axonal projection patterns at the level of DR serotonin neuronal populations that include multiple transcriptomic clusters. The data and tools we generated will enable future testing of whether a more precise correspondence exists at the level of single transcriptomic clusters that we have defined here. Our transcriptome data suggest that each DR/MR serotonin neuron type can be distinguished from others by the expression of two marker genes (*Figure 1D*; *Figure 3—figure supplement 1–3*), supporting the view that neuronal subtypes are generally specified by unique combination of genes rather than single genes (*Li et al., 2017*). With the generation of drivers based on these marker genes, intersectional methods in combination with location-specific viral targeting can be used in the future to dissect the projection patterns of the individual transcriptomic clusters.

## Insights from single-cell reconstruction

The complexity of whole-brain projection patterns of *Vglut3*⁺ and *Trh*⁺ populations discussed above can be driven by: (1) heterogeneity of projection patterns of different transcriptomic clusters within the *Vglut3*⁺ or *Trh*⁺ population; (2) heterogeneity of projection patterns of serotonin neurons within the same transcriptomic cluster; (3) complex collateralization patterns of individual serotonin neurons; and 4) a combination of some or all of the above. If scenario one were the only contributing factor, then there would be only six different projection patterns for the pDR serotonin neurons. However, our single-cell reconstruction of DR serotonin neurons revealed many more than six branching patterns (e.g., *Figure 8G*), indicating that there must be diverse collateralization patterns even within the same transcriptomic cluster, and highlighting the complexity of individual serotonin neuron projections. Despite the complexity, these single-cell tracing data nevertheless suggest certain rules obeyed by serotonin neurons.

First, there is a general segregation of cortical- and subcortical-projecting serotonin neurons. Of the 46 forebrain-projecting DR serotonin neurons, 34 have a strong preference (>80% forebrain-projecting axon length) for innervating either cortical or subcortical regions. This is likely an underestimate of the preference, especially for cortical-projecting ones, as their axons necessarily need to travel through the subcortical regions to reach the cortex. (As a specific example, most forebrain-projecting DR serotonin neurons pass through the lateral hypothalamus to reach their targets; it is thus difficult to determine whether axons in the lateral hypothalamus play a local function.) Second, most of the serotonin neurons tend to focus a majority of their arborization within one brain region (e.g., prefrontal vs. entorhinal cortex, *Figure 8B*; CeA vs. cortical amygdala (CoA), *Figure 8—figure supplement 4A₃*). The subcortical-projecting serotonin neurons appear to have more specificity, with most neurons exhibiting dense arborization in one or two nuclei. The cortical-projecting serotonin neurons can have elaborate arborization patterns across multiple cortical areas (e.g., *Figure 8—figure supplement 1C*) and the longest total axon lengths per cell (*Figure 8H*).

Together with our study on the projection-defined serotonin sub-systems (*Ren et al., 2018*), we believe it is unlikely that the major function of the forebrain-projecting serotonin system is to broadcast information non-discriminately. We note that a previous study has reconstructed axonal projections of sparsely-labeled DR neurons in the rat using microiontophetic injection of biotin dextran amine, followed by serial histological section and reconstruction (*Gagnon and Parent, 2014*). While the reported projection and arborization complexity of individual DR neurons appear qualitatively similar to our findings, these studies cannot be directly compared because the Gagnon and Parent study did not validate the serotonin identity of individually traced DR neurons, which could include Tph2-negative long-distance projecting GABAergic and glutamatergic neurons (*Bang and Commons, 2012*). By combining fMOST (*Gong et al., 2016*) and the dual-AAV sparse labeling system (*Lin et al., 2018*), our strategy ensures that we are precisely tracing genetically defined individual serotonin neurons and their projections across the whole brain. But our study is still limited by the scope (50 reconstructed cells out of 9000 DR serotonin neurons). To fully reveal the organizational logic of the serotonin system, efforts should be put into larger scale single-cell reconstruction integrated with molecular identity and functional studies of individual transcriptomic clusters of serotonin neurons.

## Integrating multiple features within individual serotonin sub-systems

The molecular features of specific serotonin cell types suggest their functional properties. For example, several studies have reported that subgroups of serotonin neurons in the MR and DR express Vglut3 and indeed, subsequent slice recording confirmed that serotonin terminals can co-release glutamate and serotonin (*Domonkos et al., 2016*; *Liu et al., 2014*; *Ren et al., 2018*; *Sengupta et al., 2017*; *Varga et al., 2009*; *Wang et al., 2019*). In addition to neurochemical properties, each serotonin neuron population expresses a specific combination of distinct genes responsible for electrophysiological (ion channels), connectivity (wiring molecules), and functional (neurotransmitter/peptide receptors) properties (*Figure 4*; *Figure 4—figure supplement 1A*). For example, most *Crhr2*⁺ neurons co-express *Vglut3* and *Npy2r*, which suggests that these serotonin neurons use glutamate as co-transmitter in their cortical targets, and are in turn modulated by corticotropin-releasing hormone and neuropeptide Y. Meanwhile, most *Trh*⁺ serotonin neurons co-express *Gad1*, *Kcns1*, and α1$_A$ adrenergic receptors (*Adra1a*) specifically. We can speculate that these serotonin neurons use Trh (and perhaps GABA) as co-transmitters to regulate the physiology of their subcortical targets, and are in turn modulated by locus coeruleus norepinephrine neurons.

Our previous study suggests that DR serotonin sub-systems have biased input but segregated output (*Ren et al., 2018*). Here we found that each of the transcriptomic clusters of serotonin neurons have distinct combination of axon guidance and cell adhesion molecules (*Figure 4—figure supplement 1A*). These differentially expressed wiring molecules might be used during development to set up distinct projection patterns of different serotonin neuron types (*Deneris and Gaspar, 2018*; *Kiyasova and Gaspar, 2011*; *Maddaloni et al., 2017*), and/or used in adults to maintain wiring connectivity or contribute to the remarkable ability of serotonergic axons to regenerate after injury (*Jin et al., 2016*).

In conclusion, our comprehensive transcriptomic dataset and identification of 11 distinct groups of the DR and MR serotonin neurons have revealed the molecular heterogeneity of the forebrain-projecting serotonin system. We have shown that the molecular features of these distinct serotonin

groups reflect their anatomical organization and provide tools for future exploration of the full projection map of molecularly defined serotonin groups. The molecular architecture of serotonin system lays the foundation for integrating anatomical, neurochemical, physiological, and behavioral functions. This integrated understanding of serotonin can in turn provide novel therapeutic strategies to treat brain disorders involving this important neuromodulator.

# Materials and methods

## Key resources table

| Reagent type (species) or resource | Designation | Source or reference | Identifiers | Additional information |
|---|---|---|---|---|
| Antibody | anti-TPH2 (rabbit polyclonal) | Novus | Cat# NB100-74555 | IF (1:1000) |
| Antibody | anti-GFP (chicken polyclonal) | Aves Labs Inc | Cat# GFP-1020 | IF (1:2000); iDISCO (1:1000) |
| Antibody | anti-rabbit donkey antibody conjugated with Cy3 | Jackson ImmunoResearch | Cat# 711-165-152 | 1:500 |
| Antibody | anti-rabbit donkey antibody conjugated with Cy5 | Jackson ImmunoResearch | Cat# 711-496-152 | 1:500 |
| Antibody | anti-chicken donkey antibody conjugated with Cy2 | Jackson ImmunoResearch | Cat# 703-605-155 | 1:500 |
| Recombinant DNA reagent | pAAV-Ef1a-fDIO-EYFP (plasmid) | Addgene | Cat# 27437 | |
| Recombinant DNA reagent | STOPx3 (plasmid) | Addgene | Cat# 22799 | |
| Recombinant DNA reagent | Membrane tag (plasmid) | Addgene | Cat # 71760 | |
| Commercial assay or kit | Gibson Assembly Master Mix | New England Biolabs | Cat# E2611S | |
| Commercial assay, kit | Papain Dissociation System | Worthington | Cat# LK003150 | |
| Commercial assay, kit | C1ql2 | Molecular Instruments | NM_207233.1 | B1 amplifier v3.0 probe |
| Commercial assay, kit | Crhr2 | Molecular Instruments | NM_001288620.1 | B1 amplifier v3.0 probe |
| Commercial assay, kit | Dlk1 | Molecular Instruments | NM_001190703.1 | B2 amplifier v3.0 probe |
| Commercial assay, kit | Ret | Molecular Instruments | NM_009050.2 | B4 amplifier v3.0 probe |
| Commercial assay, kit | Gad1 | Molecular Instruments | NM_008077.5 | B1 amplifier v3.0 probe |
| Commercial assay, kit | Gad2 | Molecular Instruments | NM_008078.2 | B1 amplifier v3.0 probe |
| Commercial assay, kit | Spp1 | Molecular Instruments | NM_001204202.1 | B4 amplifier v3.0 probe |
| Commercial assay, kit | Syt2 | Molecular Instruments | NM_001355726.1 | B1 amplifier v3.0 probe |
| Commercial assay, kit | Tacr3 | Molecular Instruments | NM_021382.6 | B1 amplifier v3.0 probe |
| Commercial assay, kit | Irx2 | Molecular Instruments | NM_010574.4 | B1 amplifier v3.0 probe |
| Commercial assay, kit | Npas1 | Molecular Instruments | NM_008718.2 | B4 amplifier v3.0 probe |

*Continued on next page*

*Continued*

| Reagent type (species) or resource | Designation | Source or reference | Identifiers | Additional information |
|---|---|---|---|---|
| Commercial assay, kit | Piezo2 | Molecular Instruments | NM_001039485.4 | B1 amplifier v3.0 probe |
| Commercial assay, kit | Tpbg | Molecular Instruments | NM_001164792.1 | B4 amplifier v3.0 probe |
| Commercial assay, kit | Met | Molecular Instruments | NM_008591.2 | B4 amplifier v3.0 probe |
| Commercial assay, kit | Trh | Molecular Instruments | NM_009426.3 | B2 amplifier v3.0 probe |
| Commercial assay, kit | Vglut3 | Molecular Instruments | NM_001310710.1 | B4 amplifier v3.0 probe |
| Commercial assay, kit | TPH2 | Molecular Instruments | NM_173391 | B3 amplifier v3.0 probe |
| Chemical compound, drug | Tetrodotoxin | Tocris Bioscience | Cat# 1069 | 1 µM |
| Chemical compound, drug | Kynurenic acid | Millipore Sigma | Cat# K3375 | 500 µM |
| Chemical compound, drug | D-AP5 | Tocris Bioscience | Cat# 0106 | 50 µM |
| Chemical compound, drug | Actinomycin D | Sigma | Cat# A1410 | 50 µM |
| Software, algorithm | IMARIS | Bitplane | RRID:SCR_007370 | Bitplane.com |
| Software, algorithm | Ilastik | GNU General Public License | RRID:SCR_015246 | https://ilastik.org/ |
| Software, algorithm | Elastix | Image Sciences Institute | RRID:SCR_009619 | https://elastix.isi.uu.nl/ |
| Software, algorithm | MATLAB | Mathworks | RRID:SCR_001622 | Mathworks.com |
| Software, algorithm | Fiji | PMID: 22743772 | RRID:SCR_002285 | https://imagej.net/Fiji |
| Software, algorithm | Allen Institute's Common Coordinate Framework (CCF) | Allen Institute for Brain Science (https://www.alleninstitute.org/) | | https://download.alleninstitute.org/informatics-archive/current-release/mouse_ccf/ |
| Software | R v3.5.3 | R-project | RRID:SCR_001905 | |
| Software | Seurat v3.0 | https://github.com/satijalab/seurat | RRID: SCR_016341 | *Butler et al., 2018*; *Stuart et al., 2019* |
| Software | STAR 2.6.1a | https://github.com/alexdobin/STAR | RRID:SCR_015899 | *Dobin et al., 2013* |
| Software | HTseq 0.11.2 | European Molecular Biology Laboratory | RRID:SCR_005514 | *Anders et al., 2015* |
| Other | Normal donkey serum | Jackson ImmunoResearch | Cat# 017-000-121 | |
| Other | DAPI stain | Invitrogen | Cat# D1306 | (1 µg/mL) |
| Other | SH800S FACS sorter | SONY | | |
| Other | NextSeq500 | Illumina | https://www.illumina.com | |

## Animals

All procedures followed animal care and biosafety guidelines approved by Stanford University's Administrative Panel on Laboratory Animal Care and Administrative Panel of Biosafety in accordance with NIH guidelines. For scRNA-seq (*Figures 1* and *4*), eight male and six female mice aged 40–45 days on a C57BL/6J background were used. The Ai14 tdTomato Cre reporter mice (JAX Strain 7914)

and *Sert-Cre* (MMRRC, Stock #017260-UCD) were crossed and the offspring was used where indi-cated. For HCR experiments (*Figure 2*) wild-type male and female mice aged 8 weeks on a C57BL/6J background were used. For whole brain axon tracing experiments (*Figures 5–7*), male and female mice aged 8–16 weeks on a CD1 and C57BL/6J mixed background were used. The *Vglut3-Cre* (also known as *Slc18a8-Cre*; JAX Strain 18147), *Thr-ires-Cre* (JAX Strain 032468; gift from Dr. Bradford B. Lowell), *Sert-Flp* (generated in this study; JAX Strain 034050), *H11-CAG-FRT-stop-FRT-EGFP* (gener-ated in this study; JAX Strain 034051), *IS* (JAX Strain 028582; gift from Dr. Z. Josh Huang) were used where indicated. For single-cell reconstruction experiments, male and female mice aged 6–16 weeks on a CD1 and C57BL/6J mixed background were used. Mice were group-housed in plastic cages with disposable bedding on a 12 hr light/dark cycle with food and water available ad libitum.

## Generation of *Sert-Flp* mice

*Sert-Flp* was generated by the Gene Targeting and Transgenics core at Janelia Research Campus.It was generated by homologous recombination in embryonic stem cells using standard procedures. A cassette of *IRES-FlpO-loxP2272-ACE-Cre POII NeoR-loxp2272* was inserted after the TAA stop codon of *Sert.* Targeting was verified in embryonic stem cells by long-arm PCR. After microinjection, chimaeras were bred with CD-1 females and F1 offspring were screened by long-arm PCR to identify mice with germline transmission of the correctly targeted construct. As previously described for the *Sert-Cre* line, the *Sert-Flp* line appears to drive transgenic reporter expression outside the raphe serotonin neurons (including neurons in the retina, cortex, and thalamus) because of the transient expression of *Sert* during development (*Lebrand et al., 1998*).

## Generation of *H11-CAG-FRT-stop-FRT-EGFP* mice

*H11-CAG-FRT-stop-FRT-EGFP* was generated using site-specific integrase-mediated transgenesis via pronuclear injection by Stanford transgenic facility. The *CAG-FRT-Stop-FRT-EGFP* transgene plasmid along with φC31 integrase RNA into the cytoplasm of mouse (FVB) embryos carrying three attP sites at *H11* locus was injected (*Tasic et al., 2011*). The transgene plasmid was constructed with pBT378.3 as the backbone. The *pCA-GFP (PACI-ASCI)* fragment was replaced with a caseate con-taining *pCA-frt neo-stop frt::GFP4m-sv40 polyA*. Cloning junctions were confirmed by sanger sequence. The integration in the H11 P3 locus was confirmed by PCR with primer pairs containing 425 ggtgataggtggcaagtggtattc, 436 atcaactaccgccacctcgac, 522 cgatgtaggtcacggtctcg, 387 gtgggactgcttttccaga.

## **Transcriptome analysis**

### Single-cell isolation and sequencing

Lysis plates were prepared by dispensing 4 µl lysis buffer as described in *Schaum et al. (2018)*. After dissociation, single tdTomato$^+$ cells were sorted in 96-well plates using SH800S (Sony). Immediately after sorting, plates were sealed with a pre-labelled aluminum seal, centrifuged, and flash frozen on dry ice. cDNA synthesis and library preparation were performed using the Smart-seq2 protocol (*Picelli et al., 2014*). Wells of each library plate were pooled using a Mosquito liquid handler (TTP Labtech). Pooling was followed by two purifications using 0.7x AMPure beads (Fisher, A63881). Library quality was assessed using capillary electrophoresis on a Fragment Analyzer (AATI), and libraries were quantified by qPCR (Kapa Biosystems, KK4923) on a CFX96 Touch Real-Time PCR Detection System (Bio-Rad). Libraries were sequenced on the NextSeq 500 Sequencing System (Illu-mina) using 2 × 75 bp paired-end reads and 2 × 8 bp index reads.

### Data processing

Sequences from the NextSeq were de-multiplexed using bcl2fastq version 2.19.0.316. Reads were aligned to the mm10 genome using STAR version 2.6.1a (*Dobin et al., 2013*). Gene counts were produced using HTseq version 0.11.2 (*Anders et al., 2015*) with default parameters, except 'stranded' was set to 'false', and 'mode' was set to 'intersection-nonempty'. Genes located on Y chromosome were removed from the count table to exclude sex bias.

## Clustering

Standard procedures for filtering, variable gene selection, dimensionality reduction and clustering were performed using the Seurat package version 3.0 (*Butler et al., 2018*; *Stuart et al., 2019*). Specifically, cells with fewer than 300 detected genes were excluded. A gene was counted as expressed if it has at least one read mapping to it and is detected in at least 3 cells. Cells with fewer than 50,000 reads were excluded. Counts were log-normalized for each cell using the natural logarithm of 1 + counts per million [ln(CPM+1)]. All genes were projected onto a low-dimensional subspace using principal component analysis. Cells were clustered using a variant of the Louvain method that includes a resolution parameter in the modularity function (*Schaum et al., 2018*). Specifically, cells were clustered based on their PCA scores. To define the number of PCs to use we performed a resampling test (JackStraw test, similarly to *Macosko et al., 2015*) and used a –value cut-off of 0.005 to choose the 'significant' PCs for the downstream analysis. To cluster the cells, we next used Louvain algorithm to iteratively group cells together, with the goal of optimizing the standard modularity function. (FindClusters functions in Seurat with the following parameters: dims = 1:20, resolution = 1).

Cells were visualized using a 2-dimensional t-distributed Stochastic Neighbour Embedding (tSNE) of the PC-projected data. Molecularly distinct cell populations were assigned to each cluster based on differentially expressed genes. Plots showing the expression of the markers for each cell subtype appear in the *Figure 3—figure supplement 1–3*.

## Gene co-expression networks

The relationship between gene expression was measured using rank correlation statistics (*Supplementary file 4*). Pearson correlations were computed across all cells. We first removed low expressed genes by selecting genes with mean expression CPM > 2, leaving ~8000 genes in the dataset. Pearson correlation coefficients ($r_p$) were computed for each gene and significance was tested by bootstrapping (1000 iterations). A correlation table containing $r_p$ above 0.3 and below –0.3 can be found in *Supplementary file 3*. Reported values are mean from the bootstrapped values. Gene functional categories were retrieved from HGNC resource (https://www.genenames.org). Genes assigned to more than one functional category were re-assigned a single category in the following priority order: SNAREs, Secreted Ligands, ACh and Monoamine Receptors, Glutamate Receptors, Axon Guidance and Cell Adhesion Molecules (CAMs), GABA Receptors, GPCRs, Ion Channel Proteins, Transcription Factors, a full list of genes assigned to each category can be found in *Supplementary file 2*. Gene pairs for which $r_p$ <0.4 were removed and remaining pairs were visualized as a network using *igraph* and *visNetwork* R packages. To further refine the final list of co-expressed genes and generate *Figure 4D* we focused on gene pairs for which:1) $r_p$ >0.5; 2) at least one gene of the pair is found among cluster markers; 3) both genes of the correlating pair belong to one of the above listed functional categories.

## Data availability

The datasets generated and analyzed in the study are available in the NCBI Gene Expression Omnibus (GSE135132, https://www.ncbi.nlm.nih.gov/geo/query/acc.cgi?acc=GSE135132).

## Abbreviations for anatomical regions

| | |
|---|---|
| ACA | anterior cingulate area |
| AHN | anterior hypothalamic nucleus |
| aHy | anterior hypothalamic regions |
| AI | anterior insular cortex |
| AOB | accessory olfactory bulb |
| AON | anterior olfactory nucleus |
| AUD | auditory cortex anterior |
| aVIS | visual cortex |

*Continued on next page*

| | |
|---|---|
| BLA | basolateral amygdala |
| BNST | bed nucleus of the stria terminalis |
| BST | nucleus of stria terminalis |
| CA | cornu amonis |
| CB | cerebellum |
| cBS | caudal brainstem |
| CeA | central amygdala |
| CLI | central linear nucleus raphe |
| CoA | cortical amygdala |
| Col | colliculus |
| cMid | caudal midbrain |
| DB | nucleus of the diagonal band |
| DCN | deep cerebellum nuclei |
| DG | dentate gyrus |
| dLGN | dorsal lateral geniculate nucleus |
| dMid | dorsal midbrain |
| DMH | dorsomedial nucleus of the hypothalamus |
| DP | dorsal peduncular cortex |
| dPED | dorsal peduncular area |
| DR | dorsal raphe nucleus |
| dStr | dorsal striatum |
| ECT | ectorhinal cortex |
| ENT | entorhinal cortex |
| EP | endopiriform nucleus |
| GU | gustatory area |
| HY | hypothalamus |
| ILA | infralimbic area |
| LHb | lateral habenula |
| LHy | lateral hypothalamus |
| LS | lateral septal nucleus |
| LS | lateral septal nucleus |
| M1 | primary motor area |
| MeA | medial amygdala |
| MG | medial geniculate nucleus |
| MOB | main olfactory bulb |
| MR | median raphe nucleus |
| MD | medial dorsal nucleus of the thalamus |
| NLOT | nucleus of lateral olfactory tract |
| NST | nucleus of solitary tract |
| OA | olfactory area |
| OB | olfactory bulb |
| OFC | orbitofrontal cortex |
| ORB | orbital area |
| oHPC | other hippocampal regions |
| PAA | piriform – amygdala area |

| | |
|---|---|
| pAAs | posterior parietal association areas |
| PERI | perirhinal cortex |
| pHy | posterior hypothalamic regions |
| PIR | piriform cortex |
| PL | prelimbic area |
| pPIR | posterior piriform cortex |
| PSTh | parasubthalamic nucleus |
| PTLP | posterior parietal association area |
| PVH | paraventricular hypothalamus |
| PVHd | paraventricular hypothalamus, descending division. |
| RSP | retrosplenial area |
| S1-B | primary somatosensory area, barrel field |
| SI | substantia innominata |
| SMT | submedial nucleus of the thalamus |
| SNc | substantia nigra compacta |
| SNr | substantia nigra pars reticulata |
| SS | somatosensory |
| Sth | subthalamic nucleus |
| STR | striatum |
| TEA | temporal association |
| TH | thalamus |
| TR | piriform transition area |
| TT | tenia tecta |
| VIS | visual cortex |
| VLPO | ventrolateral preoptic nucleus |
| vmHy | ventromedial hypothalamic regions |
| vMid | ventral midbrain |
| vStr | ventral striatum |
| ZI | zona incerta |

## Stereotaxic surgeries

Mice were anesthetized either with 1.5–2.0% isoflurane and placed in a stereotaxic apparatus (Kopf Instruments). For virus injection in to the DR, the following coordinates (in mm) were used: –4.3AP, 1.10 ML, –2.85 DV; –4.3AP, 1.10 ML, –2.70 DV, with 20° ML angle. (AP is relative to bregma; DV is relative to the brain surface when AP is –1.0). After surgery, mice recovered on a heated pad until ambulatory and then returned to their homecage.

## Viral constructs

The full design of Ef1a-CreON/FlpON-mGFP is *Ef1a-fDIO-[membrane tag]-Kozak-loxP-STOPx3-loxP-ΔGFP*. The AAV vector backbones that contained the *Ef1a-fDIO were derived from pAAV-Ef1a-fDIO-EYFP* (Addgene, #27437) (*Fenno et al., 2014*). The *[membrane tag]-Kozak-loxP* sequence was synthesized by GenScript. The membrane tag was the N-terminal 20 amino acids of Gap43 (Addgene, #71760). The sequence of *loxP-ΔGFP* were cloned by PCR and *-ΔGFP is GFP sequence omitting ATG*. And then these two pieces together with *STOPx3* (Addgene, #22799) were ligated into the *AAV-Ef1a-fDIO* backbone in the antisense orientation by Gibson assembly. DNA oligonucleotides were synthesized by Elim Biopharmaceuticals Inc and GenScript.

## Viruses packaging

For whole brain tracing, the AAV vector carrying *Ef1a-CreON/FlpON-mGFP* were packaged into AAV2/8 serotype with $1 \times 10^{12}$ gc/ml by Gene Vector and Virus Core, Stanford University. 500 nl of *AAV-CreON/FlpON-mGFP* was injected into the DR for each mouse. For single cell reconstruction, AAV vectors carrying the *TRE-DIO-FlpO*, *TRE-fDIO-GFP-IRES-tTA* construct were packaged into AAV2/9 serotype with titres $10^9$–$10^{10}$ gc/ml in Dr. Minmin Luo's lab as described before (*Lin et al., 2018*). *AAV-TRE-DIO-FLPo* ($10^7$ gc/ml) virus and *AAV-TRE-fDIO-GFP-IRES-tTA* ($10^9$ gc/ml) virus were mixed with the ratio of 1:9. 200 nL mixed virus was injected into the DR for each mouse.

## Histology and imaging

Animals were perfused transcardially with phosphate buffered saline (PBS) followed by 4% paraformaldehyde (PFA). Brains were dissected, post-fixed in 4% PFA for 12–24 hr in 4°C, then placed in 30% sucrose for 24–48 hr. They were then embedded in Optimum Cutting Temperature (OCT, Tissue Tek) and stored at–80°C until sectioning. For the antibody staining in *Figures 1*, 50-μm sections containing DR were collected onto Superfrost Plus slides to maintain the anterior to posterior sequence. All working solutions listed below included 0.2% $NaN_3$ to prevent microbial growth. Slides were then washed $3 \times 10$ min in PBS and pretreated overnight with 0.5 mM SDS at 37°C. Slides were then blocked for 4 hr at room temperature in 10% normal donkey serum (NDS) in PBS with 0.3% Triton-X100 (PBST), followed by incubation in primary antibody (Novus, rabbit anti-Tph2) diluted 1:1000 in 5% NDS in PBST for 24 hr at RT. After $3 \times 10$ min washes in PBS, secondary antibody was applied for 6 hr at room temperature (donkey anti-rabbit, Alexa-647 or Alexa-488, Jackson ImmunoResearch), followed by $3 \times 10$ min washes in PBST. Slides were then stained for NeuroTrace Blue (NTB, Invitrogen). For NTB staining, slides were washed $1 \times 5$ min in PBS, $2 \times 10$ min in PBST, incubated for 2–3 hr at room temperature in (1:500) NTB in PBST, washed $1 \times 20$ min with PBST, and $1 \times 5$ min with PBS. Sections were additionally stained with DAPI (1:10,000 of 5 mg/mL, Sigma-Aldrich) in PBS for 10–15 min and washed once more with PBS. Slides were mounted and coversliped with Fluorogel (Electron Microscopy Sciences). After that, the slides were then imaged either using a Zeiss 780 confocal microscope or a 3i spinning disk confocal microscope (CSU-W1 SoRa), and images were processed using NIH ImageJ software. After that, whole slides were then imaged with a 5x objective using a Leica Ariol slide scanner with the SL200 slide loader.

For DR-containing slices in *Figures 5* and *6*, staining was applied to floating sections. Primary antibodies (Novus, rabbit anti-Tph2, 1:1000; Aves Labs Inc, chicken anti-GFP, 1:2000) were applied for 48 hr and secondary antibodies for 12 hr at 4°C. For serotonin terminal staining in *Figure 6*, floating sections were stained with Primary antibodies (Aves Labs Inc, chicken anti-GFP, 1:2000) for 60 hr and secondary antibodies for 18 hr at 4°C.

## Whole brain imaging of *Vglut3*$^+$ and *Trh*$^+$ serotonin projections

After 8–10 weeks of virus expression, mice were transcardially perfused with 20 ml 1x PBS containing 10 μg/ml heparin followed by 20 ml of 4% PFA before removing each brain and allowing them to postfix overnight at 4° C. The clearing protocol largely follows the steps outlined in *Chi et al. (2018)*. Brains were washed at room temperature with motion at least 30 min each step: 3x in 1x PBS before switching to 0.1% Triton X-100 with 0.3 M glycine (B1N buffer) and being stepwise (20/40/60/80%) dehydrated into 100% methanol. Delipidation was carried out by an overnight incubation in 2:1 mixture of DCM:methanol and a 1 hr incubation in 100% DCM the following day. After 3x washes in 100% methanol, brains were bleached for 4 hr in a 5:1 mix of methanol: 30% $H_2O_2$ and then stepwise (80/60/40/20%) restored into B1N buffer. Samples were permeabilized with 2 washes of PTxwH buffer containing 5% DMSO and 0.3M glycine for 3 hr before being washed in PTxwH overnight. Antibody labeling was carried out in PTxwH buffer at 37° C with motion. Primary antibody (Aves 1020; chicken anti-GFP, 1:1000) was added and samples were incubated for 11 days. After 3 days of washes, secondary antibody (Thermo A-31573, AlexaFluor 647 donkey anti-chicken, 1:1000) was added for 8 days, followed by another 3 days of washes. One final day of washing in 1x PBS preceded clearing. Samples were again dehydrated stepwise into methanol, using water as the alternative fraction. Delipidation proceeded as before with a DCM/methanol mixture overnight and $2 \times 1$ hr DCM-only incubations the next day. Brains were finally cleared for 4 hr in dibenzyl ether and then stored in a fresh tube of dibenzyl ether at least 24 hr before imaging.

Samples were imaged with a LaVision Ultramicroscope II lightsheet using a 2x objective and 3 µm z-step size. Antibody fluorescense was collected from a 640 nm laser and autofluorescense captured from 488 nm illumination. Image volumes were processed and analyzed with custom Python and MATLAB scripts. In short, we trained a 3D U-Net convolutional neural network (*Çiçek et al., 2016*) to identify axons in volumes and post-processed the resulting probability-based volumes as previously described (*Ren et al., 2018*). Python code for training is available at GitHub (https://github.com/AlbertPun/TRAILMAP; copy archived at https://github.com/elifesciences-publications/TRAILMAP) (*Friedmann et al., 2019*). Using the autofluorescent channel, we aligned samples to the Allen Institute's Common Coordinate Framework (*Renier et al., 2016*), applied the same transformation vectors to the volumetric projection of axons, and quantified total axon content in each brain region listed in *Supplementary file 5*.

## Hybridization chain reaction in situs

Probes were generated for use with HCR v3.0 (Molecular Technologies) (*Choi et al., 2018*). Wildtype 8 week old mice were perfused and brains were removed and fixed as described above. Following an overnight postfix, brains were cryoprotected in 30% sucrose until they sank and subsequently frozen at −80° C. The midbrain raphe nuclei were sectioned coronally at either 16 or 20 µm directly onto a glass slide, and dried at room temperature for 4–6 hr before storing at −20° C overnight. Sections were re-fixed again in 4% PFA, washed in 1x PBS, treated with a solution containing 50 ml $H_2O$, 0.5 ml 1M Tris ph7.4, 100 µl 0.5M EDTA, 35 µl ProK for 6 min at 37° C before another round of PFA fixation and PBS washing. Hybridization and amplification steps were carried out as recommended by Molecular Technologies. Imaging was performed with a Zeiss 780 LSM confocal using a 20x objective. Double labeled cells were counted manually using the CellCounter plugin for FIJI software.

## Single Cell Reconstruction

### fMOST imaging and image preprocessing

4–6 weeks after virus injection, brains were dissected and post-fixed in 4% paraformaldehyde for 24 hr at 4°C. The brains were rinsed in 0.01 M PBS (Sigma-Aldrich) three times (for 2 hr each) and embedded in Lowicryl HM20 resin (Electron Microscopy Sciences, 14340). We use a fluorescence micro-optical sectioning tomography (fMOST) system to acquire the brain-wide image dataset at high resolutions (0.23 × 0.23 × 1 µm for 10 brains and 0.35 × 0.35 × 1 µm for the other nine brains). Embedded brain samples were mounted on a 3D translation stage in a water bath with propidium iodide (PI). The fMOST system automatically performs the coronal sectioning with 1 um steps and imaging with two color channels in 16-bit depth. The green channel of GFP is for visualization of neurons and the red channel of PI counterstaining is for visualization the whole brain cytoarchitecture.

### Image annotation and skeletonization

Amira software (v 5.4.1, Visage imaging, Inc) were used for semi-automatically reconstruction of single neurons. First, we use Amira to load a relatively large volume but low-resolution data and find the position of soma or major axon as a start position. Then, one by one, we load each small volume (800 × 800 × 400 voxels) of highest resolution data along the axons and dendrites to label the full structure of each neuron. We have totally reconstructed 50 high quality neurons with complete and clear axon terminals. The reconstructed neurons were checked by one another person.

### Image registration and visualization

Reconstructed neurons were aligned to the Allen Common Coordinate Framework (CCF). Elastix (Klein 2010) were used to register the manually segmented outline of a downsampled version of the PI-staining vloume (25 × 25 × 25 µm voxel size) to the average template of CCF (25 × 25 × 25 µm voxel size). An initial affine registration was applied for align the sample to CCF, followed by an iterative non-rigid b-spline transformation for more precise registration. The generated parameters were used for coordinate transformation of reconstructed neurons to align them to the CCF. Cinema 4d (MAXON Computer GmbH, Germany) and customized python scripts were used to generate 3D rendered images and movies.

## Data analysis and quantification

We use customized program in MATLAB (MathWorks) for the fiber lengths calculation and heatmap generation. The projection diagram was drawn with Microsoft Office Visio.

## Acknowledgements

We thank Caiying Guo and the Howard Hughes Medical Institute/Janelia Research Campus for assisting the generation of the *Sert-Flp* mice, Yanru Chen and the Stanford transgenic core for assisting the generation of the *H11-CAG-FRT-stop-FRT-EGFP* mice, Bradford B Lowell for *Thr-ires-Cre* mice, Z Josh Huang for the *IS* mice, Stanford Gene Vector and Virus Core for producing virus, Kang Shen and Nathan McDonald for sharing and advising the 3i spinning disc confocal microscope. Norma Neff and Jennifer Okamoto for sequencing expertise. Zhiyong Guo for assisting several cases of single cell reconstruction, Yanwen Sun for assisting the generation of a customized program related to 3D images processing, Tie-Mei Li for assisting the cloning of *Ef1a-CreON/FlpON-mGFP*.

## Additional information

### Competing interests

Jennifer L Raymond: Reviewing editor, *eLife*. The other authors declare that no competing interests exist.

### Funding

| Funder | Grant reference number | Author |
|---|---|---|
| National Institutes of Health | R01 NS104698 | Liqun Luo |
| National Science Foundation | NeuroNex | Liqun Luo |
| National Institutes of Health | NS057488 | Jennifer L Raymond |
| Ministry of Science and Technology of the People's Republic of China | 2015BAI08B02 | Minmin Luo |
| National Natural Science Foundation of China | 91432114 | Minmin Luo |
| National Natural Science Foundation of China | 91632302 | Minmin Luo |
| Beijing Municipal Government | | Minmin Luo |
| NSFC | 61721092 | Qingming Luo |
| NSFC | 81827901 | Qingming Luo |
| Swiss National Science Foundation | PostDoc Mobility Fellowship | Alina Isakova |

The funders had no role in study design, data collection and interpretation, or the decision to submit the work for publication.

### Author contributions

Jing Ren, Conceptualization, Data curation, Formal analysis, Validation, Methodology, Writing—original draft, Writing—review and editing, Collected single-cell samples, Developed and characterized *Sert-Flp* mice and intersectional viral-genetic strategies, Performed whole-brain intersectional axon tracing experiments with assistance from SG, Analyzed the single-cell reconstruction data; Alina Isakova, Data curation, Formal analysis, Visualization, Methodology, Writing—review and editing, Collected single cell samples, Processed the cells for single-cell RNA-seq, Performed scRNA-seq data analysis with support from SRQ; Drew Friedmann, Data curation, Software, Formal analysis, Validation, Visualization, Methodology, Writing—review and editing, Performed FISH experiments and analyzed data with assistance from SG, Performed whole-brain

intersectional axon tracing experiments with assistance from SG, Analyzed whole-brain axon maps using algorithms originally developed by AP, Analyzed the single-cell reconstruction data; Jiawei Zeng, Data curation, Visualization, Writing—review and editing, Performed the bulk of single axon tracing with assistance from RW and RL and support from ML, Analyzed the single-cell reconstruction data; Sophie M Grutzner, Data curation, Validation, Assisted with performing FISH experiments and analyzing data, Assisted with performing whole-brain intersectional axon tracing experiments; Albert Pun, Software; Grace Q Zhao, Qingming Luo, Resources; Sai Saroja Kolluru, Ruiyu Wang, Rui Lin, Pengcheng Li, Anan Li, Data curation; Jennifer L Raymond, Minmin Luo, Stephen R Quake, Resources, Writing—review and editing; Liqun Luo, Conceptualization, Resources, Supervision, Funding acquisition, Writing—original draft, Writing—review and editing

### Author ORCIDs
Jing Ren (iD) https://orcid.org/0000-0002-6665-7457
Alina Isakova (iD) https://orcid.org/0000-0003-1113-6889
Jennifer L Raymond (iD) http://orcid.org/0000-0002-8145-747X
Qingming Luo (iD) http://orcid.org/0000-0002-6725-9311
Minmin Luo (iD) http://orcid.org/0000-0003-3535-6624
Stephen R Quake (iD) http://orcid.org/0000-0002-1613-0809
Liqun Luo (iD) https://orcid.org/0000-0001-5467-9264

### Ethics
Animal experimentation: All procedures followed animal care and biosafety guidelines approved by Stanford University's Administrative Panel on Laboratory Animal Care and Administrative Panel of Biosafety in accordance with NIH guidelines. The animal protocol is #14007 and the biosafety protocol is APB 2500.

### Decision letter and Author response
Decision letter https://doi.org/10.7554/eLife.49424.042
Author response https://doi.org/10.7554/eLife.49424.043

## Additional files

### Supplementary files
• Supplementary file 1. Raw data for single-cell gene expression. Column names denote cell index, row names denote gene names.
DOI: https://doi.org/10.7554/eLife.49424.033

• Supplementary file 2. Functional gene categories, used to generate *Figure 4* and *Supplementary file 4*.
DOI: https://doi.org/10.7554/eLife.49424.034

• Supplementary file 3. Pearson correlation coefficient ($r_p$) of pairwise correlation of gene expression across 999 cells. Contains gene pairs with $r_p > 0.3$, $r_p < –0.3$. Contains gene pairs with $rp > 0.3$, $rp < –0.3$.
DOI: https://doi.org/10.7554/eLife.49424.035

• Supplementary file 4. Co-expression networks. Networks were constructed based on Pearson correlation coefficient ($r_p$) of gene expression across all cells. Genes appear connected if $r_p > 0.4$. Edge width represents $r_p$. Nodes are colored according to functional gene categories (Materials and methods).
DOI: https://doi.org/10.7554/eLife.49424.036

• Supplementary file 5. Allen Brain Atlas IDs and Their Corresponding Names as Identified by the 2017 Common Coordinate Framework, Related to *Figure 7*. Regions were selected prior to analysis such that areas defined by individual layers (e.g., cortical layers I–VI), cell identity, and anatomical cardinal directions are collapsed into their parent region. Individual normalized regional densities for each brain are aligned to the heat maps from *Figure 7E*.
DOI: https://doi.org/10.7554/eLife.49424.037

• Transparent reporting form DOI: https://doi.org/10.7554/eLife.49424.038

## Data availability

Sequencing data have been deposited in GEO (GSE135132; https://www.ncbi.nlm.nih.gov/geo/query/acc.cgi?acc=GSE135132).

The following dataset was generated:

| Author(s) | Year | Dataset title | Dataset URL | Database and Identifier |
|---|---|---|---|---|
| Alina Isakova | 2019 | Single-Cell Transcriptomes and Whole-Brain Projections of Serotonin Neurons in the Mouse Dorsal and Median Raphe Nuclei | https://www.ncbi.nlm.nih.gov/geo/query/acc.cgi?acc=GSE135132 | NCBI Gene Expression Omnibus, GSE135132 |

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
