## [Decision Letter]

Thank you for submitting your article "Single-cell transcriptomes and whole-brain projections of serotonin neurons in the dorsal and medial raphe nuclei" for consideration by *eLife*. Your article has been reviewed by three peer reviewers, and the evaluation has been overseen by a Reviewing Editor and Eve Marder as the Senior Editor. The following individuals involved in review of your submission have agreed to reveal their identity: Patricia Gaspar (Reviewer #1), Sacha Nelson (Reviewing Editor).

The reviewers have discussed the reviews with one another and the Reviewing Editor has drafted this decision to help you prepare a revised submission.

Summary:

The authors use an impressive array of genomic, genetic and anatomical techniques to further illuminate the heterogeneity of serotonin neuron subtypes in the mouse. The paper includes a large number of single axon reconstructions which, together with the transcriptional data, are likely to form a valuable resource for others working in this field. The paper reports multiple new findings including new subtype-specific markers within the B7-B8 group and interesting differences between male and female animals. Some of these new findings need some additional support, but they add to enthusiasm for this detailed and systematic study.

Essential revisions:

1) The authors need to better integrate their data with those already available for the parsing of serotonin raphe neurons. Reviewers agreed that it would be helpful to relate the expression data to previously published RNAseq data including the rhombomere derived analysis from Okaty et al. A supplementary figure showing key similarities and differences will be important for the field. Alternatively or additionally, a discussion paragraph would also help.

2) Further details of the anatomical projections should be clarified. The authors do not mention the projections from the Vglut3^+^ neurons to the hippocampus, although these are clearly visible on the figures. It would be nice to show more precisely the starting point of the 3 cases analyzed for each genotype. Subsection “Whole-brain axonal arborization patterns of individual serotonin neurons”: projections from the DR to the spinal cord are mentioned (subsection “Other groups”) but no projections are shown beyond the pons. Since up to now no projections to the spinal cord from B7-B8 have been demonstrated, specific illustrations are needed if the authors want to support this claim. The inclusion of an analysis of gender differences is novel and interesting. Authors should indicate more precisely the number of males and females used (Results first paragraph) and retained after quality control. One reviewer noted that: " subsection “Whole-brain axonal projections of selected serotonin neuron subpopulations” paragraph three speaks to the high variability in projection patterns from individual brains of each genotype following viral injection. Given that each genotype had only n=3 mice, and the highly variable regions in question were targeted densely "in one or two individual brains," more "n" may be warranted to ensure accurate analysis of axonal projections for each intersectionally defined subpopulation." Although reviewers agreed that it would increase the impact of the paper to include further cases to clarify the intersectional labeling, the consensus was that this was not required for the revision.

3) The new *Sert-Flp* line should be characterized in more detail in Figure 5. For example, in Figure 5B, it would be useful to see higher-magnification images of single cells co-expressing GFP and Tph2. Are there cells in cortex (e.g., L6 corticothalamic neurons) that express the transgene, as is the case for the *Sert-Cre* mouse?

4) How were the 11 clusters defined? The authors should provide more detail about the quantitative criteria to decide on these 11 clusters, and how well separated they are from each other.

5) In Figure 2, the expression of Sox14 should be validated in serotonin neurons using immunostaining or double in situ. Sox14 expression in R1 has previously been associated with neighboring GABAergic neuron cohorts (Lahti et al., 2016).

6) In keeping with *eLife* policy, the title should include reference to the biological preparation ("mouse", or perhaps "rodent")

---

## [Author Response]

Essential revisions:1) The authors need to better integrate their data with those already available for the parsing of serotonin raphe neurons. Reviewers agreed that it would be helpful to relate the expression data to previously published RNAseq data including the rhombomere derived analysis from Okaty et al. A supplementary figure showing key similarities and differences will be important for the field. Alternatively or additionally, a discussion paragraph would also help.

We thank the reviewers for this suggestion. Because of the different nature of the dataset (most of Okaty’s data are from bulk-RNA-seq), we could only make qualitative comparisons based on differentially expressed marker genes identified in both studies. Notably, we found an approximal correspondence between rhombomere-derived division and our adult single-cell RNA-seq-based transcriptomic clusters. We summarized the comparisons in a new supplementary figure (Figure 3—figure supplement 4), and added the following new paragraph in Discussion:

“Our results are broadly consistent with previous findings that utilized developmental origin to differentiate raphe serotonin neurons (Okaty et al., 2015). […] Comparing marker genes enriched in R1–R3-derived serotonin neurons from Okaty et al. and MR-1–MR-4 transcriptomic clusters from our current study (Figure 3—figure supplement 4) indeed suggests an approximate correspondence: R1-derived serotonin neurons correspond to our MR-1 cluster, R2 to MR-4 cluster, and R3 to both MR-2 and MR-3 clusters.”

2) Further details of the anatomical projections should be clarified. The authors do not mention the projections from the Vglut3^+^ neurons to the hippocampus, although these are clearly visible on the figures.

We suspect that the Vglut3^+^ projections to the hippocampus originate from Tph2^+^/Vglut3^+^ neurons located in MR. As Muzerelle et al., 2016 showed that, MR is the major resource of serotonin projections to the hippocampus and cDR also contribute significantly. While our injections were targeted to the core of the principal dorsal raphe, we did see some virally labeled cells in MR, and we didn’t see cells labeled in the cDR (Figure 7—figure supplement 3). We have added text to highlight the presence of these MR cells in our labeled populations.

It would be nice to show more precisely the starting point of the 3 cases analyzed for each genotype.

We have added a panel to Figure 7—figure supplement 3 with a schematic indicating the center of the cell bodies labeled in each of the 6 separate brains. We further clarified in the figure legend that the z-stacks shown in panel (A) are representative of all samples and display all labeled cell bodies and as such, well-represent both the location and extent of labeling.

Subsection “Whole-brain axonal arborization patterns of individual serotonin neurons”: projections from the DR to the spinal cord are mentioned (subsection “Other groups”) but no projections are shown beyond the pons. Since up to now no projections to the spinal cord from B7-B8 have been demonstrated, specific illustrations are needed if the authors want to support this claim.

Indeed the brain registration of single-cell-reconstruction experiments ended posteriorly at the medulla. But we found one neuron’s axon appeared to extend to the caudal-most end of our sample, thus think that this neuron (Figure 8—figure supplement 3 D, #36) projected to the spinal cord. We added a word “likely” since we did not trace the axon into the spinal cord. We cited two previous reports showing that DR serotonin neurons project to the spinal cord.

The inclusion of an analysis of gender differences is novel and interesting. Authors should indicate more precisely the number of males and females used (Results first paragraph) and retained after quality control.

We thank the review for appreciating the gender differences. We have the manuscripts as follows:

“After quality control (Materials and methods), we determined the transcriptomes of 709 cells from eight samples that include MR, pDR, and cDR, and 290 cells from six pDR-only samples (999 cells in total, comprising 581 cells from female and 418 cells from male mice).”

One reviewer noted that: " subsection “Whole-brain axonal projections of selected serotonin neuron subpopulations” paragraph three speaks to the high variability in projection patterns from individual brains of each genotype following viral injection. Given that each genotype had only n=3 mice, and the highly variable regions in question were targeted densely "in one or two individual brains," more "n" may be warranted to ensure accurate analysis of axonal projections for each intersectionally defined subpopulation." Although reviewers agreed that it would increase the impact of the paper to include further cases to clarify the intersectional labeling, the consensus was that this was not required for the revision.

We agree that more “n” would strengthen these findings, however given the duration of these experiments (~3.5 months if everything works smoothly), we appreciate reviewers’ understanding that this is not required for the revision. We believe that our main conclusion (that Vglut3^+^ and TrH^+^ populations are largely complementary to each other) is well supported despite the low *n*. Our secondary findings regarding variability focus on areas that show positive signal rather than absence of signal. We agree that findings regarding the heterogeneity within the intersectionally defined subpopulation require larger n, or ideally using transcriptomic cluster-specific intersections. We therefore added the following sentence at the end of this paragraph:

“Larger scale experiments or experiments using Cre lines that are expressed in single clusters will be required to further dissect heterogeneity within the *Vglut3^+^* or *TrH^+^* serotonin neurons."

3) The new Sert-Flp line should be characterized in more detail in Figure 5. For example, in Figure 5B, it would be useful to see higher-magnification images of single cells co-expressing GFP and Tph2. Are there cells in cortex (e.g., L6 corticothalamic neurons) that express the transgene, as is the case for the Sert-Cre mouse?

We thank the reviewer’s suggestion and we added the higher-magnification images of single cells co-expressing GFP and Tph2 as insets in Figure 5B. We also updated the figure legend accordingly. As the *Sert-Cre* mouse, the *Sert-Flp* line appears to drive reporter expression outside the raphe serotonin neurons, since we did observe labeled neurons in the retina, cortex and thalamus. We didn’t fully characterize the expression in these brain regions because it is outside the scope our study. We have initiated the process of depositing the mouse line to the Jackson Lab (and have added JAX Stock number to this as well as our new FLP reporter) so others interested in these expression patterns can characterize this further.

We added a sentence in Materials and methods section regarding this point as following:

“As previously described for the *Sert-Cre* line, the *Sert-Flp* line appears to drive transgenic reporter expression outside the raphe serotonin neurons (including neurons in the retina, cortex, and thalamus) because of the transient expression of *Sert* during development (Lebrand et al., 1998).”

4) How were the 11 clusters defined? The authors should provide more detail about the quantitative criteria to decide on these 11 clusters, and how well separated they are from each other.

To address the reviewer’s question, we have updated Data Processing section of the Materials and methods section with following:

“Cells were clustered using a variant of the Louvain method that includes a resolution parameter in the modularity function (Schaum et al., 2018). Specifically, cells were clustered based on their PCA scores. To define the number of PCs to use we performed a resampling test (JackStraw test, similarly to Macosko et al. DOI:https://doi.org/10.1016/j.cell.2015.05.002) and used a value cut-off of 0.005 to choose the “significant” PCs for the downstream analysis. To cluster the cells, we next used the Louvain algorithm to iteratively group cells together, with the goal of optimizing the standard modularity function. (FindClusters() functions in Seurat with the following parameters: dims = 1:20, resolution = 1).”

5) In Figure 2, the expression of Sox14 should be validated in serotonin neurons using immunostaining or double in situ. Sox14 expression in R1 has previously been associated with neighboring GABAergic neuron cohorts (Lahti et al., 2016).

We thank the reviewer to bring to our attention about the Lahti et al. paper, which describes Sox14 expression in GABAergic neuron cohorts during development. While the Lahti et al. paper did not show Sox14 expression in GABAergic neurons in adult, it would have been consistent with our study even if it did. In our transcriptomic data, Sox14 is expressed in Clusters 1 and 2, which are fully included in clusters that express Gad1 and Gad2 (Clusters 1–3), the most widely used markers for GABAergic neurons. So our transcriptomic data already validated a population of *Tph2^+^Sox14^+^Gad1^+^Gad2^+^* neurons that were sorted based on their expression of *Sert-Cre*, and we do not see added value of immunostaining and double in situ data. It is possible that Sox14 is additionally expressed *Tph2^–^Gad1^+^Gad2^+^* neurons, but our paper is focused exclusively on serotonin neurons so characterizing marker expression outside serotonin neurons is beyond the scope of our paper.

We also want to emphasize that despite expression of *Gad1* and *Gad2*, these neurons do not express *Vgat* (vesicular GABA transporter), and therefore are not typical GABAergic neurons. Hence our *Sox14^+^* neurons cannot be due to contamination of typical GABAergic neurons nearby.

We added the following sentence into our manuscript in the transcription factor section:

“Interestingly, *Sox14*, previously reported to be associated with GABAergic neurons in developing brain (Lahti et al., 2016), is expressed in Clusters 1 and 2 in adult serotonin neurons that co-express *Gad1* and *Gad2*.”

6) In keeping with eLife policy, the title should include reference to the biological preparation ("mouse", or perhaps "rodent")

We have changed the title as “Single-Cell Transcriptomes and Whole-Brain Projections of Serotonin Neurons in the Mouse Dorsal and Median Raphe Nuclei” accordingly.